

**The Usumacinta-Grijalva beach-ridge plain in southern Mexico:**
**a high-resolution archive of river discharge and precipitation**
Kees Nooren[1], Wim Z. Hoek[1], Tim Winkels[1], Annika Huizinga[1], Hans van der Plicht[2,3], Remke
L. van Dam[4,5,6], Sytze van Heteren[7], Manfred J. van Bergen[1], Maarten A. Prins[8], Tony Reimann[9],
Jakob Wallinga[9], Kim M. Cohen[1,7,10], Philip Minderhoud[1] and Hans Middelkoop[1]
[1]Utrecht University, Faculty of Geosciences, 3508 TC Utrecht, The Netherlands;
[2]Groningen University, Center for Isotope Research, 9747 AG Groningen, The Netherlands;
[3]Leiden University, Faculty of Archaeology, 2333 CC Leiden, The Netherlands
[4]Centro Federal de Educação Tecnológica de Minas Gerais, Department of Civil Engineering
(CEFET-MG), CEP 30510-000, Belo Horizonte, Brazil
[5]Michigan State University, Department of Earth and Environmental Sciences, East Lansing, MI
48824, United States
[6]Queensland University of Technology, Science and Engineering Faculty, Institute for Future
Environments, Brisbane, QLD 4001, Australia
[7]TNO – Geological Survey of the Netherlands, Geomodeling Department, 3508 TA Utrecht, The
Netherlands
[8]Vrije Universiteit, Faculty of Earth and Life Sciences, 1081 HV Amsterdam, the Netherlands
[9]Wageningen University, Soil Geography and Landscape Group & Netherlands Centre for
Luminescence dating, 6708 PB Wageningen, The Netherlands
[10]Deltares, Department of Applied Geology and Geophysics, 3584 CB Utrecht, The Netherlands.
*Correspondence to:* Kees Nooren (c.a.m.nooren@uu.nl)
## Abstract
The beach-ridge sequence of the Usumacinta-Grijalva delta borders a 300-km-long section of the
Southern Mexico Gulf coast. With around 500 beach ridges formed in the last 6500 years, the
sequence is unsurpassed in the world in terms of numbers of individual ridges preserved,
continuity of the record, and temporal resolution. We mapped and dated the most extensively
accreted part of the sequence, linking six phases of accretion to river-mouth reconfigurations and
constraining their ages with [14]C and OSL dating. The geomorphological and sedimentological
reconstruction relied on LiDAR data, coring transects, GPR measurements, grain-size analyses
and chemical fingerprinting of volcanic glass and pumice encountered within the beach and dune
deposits.
We demonstrate that the beach-ridge complex was formed under ample long-term fluvial
sediment supply and shorter-term wave- and aeolian modulated sediment reworking. The
abundance of fluvially supplied sand is explained by the presence of easily weatherable Los
Chocoyos ignimbrites from the ca. 84 ka eruption of Atitlán volcano (Guatemala) in the
catchment of the Usumacinta River. Autocyclic processes seem responsible for the formation of
ridge/swale couplets. Fluctuations in their periodicity (ranging from 6-19 yrs) are governed by
progradation rate, and are therefore not indicative of sea level fluctuations or variability in storm
activity. The fine sandy beach ridges are mainly swash built. Ridge elevation, however, is
strongly influenced by aeolian accretion during the time the ridge is located next to the beach.
Beach-ridge elevation is negatively correlated with progradation rate, which we relate to the
variability in sediment supply to the coastal zone, reflecting decadal-scale precipitation changes
within the river catchment. In the Southern Mexican delta plain, the coastal beach ridges
therefore appear to be excellent recorders of hinterland precipitation.



## 1 Introduction

Beach-ridge plains with long sequences holding many individual ridges consisting of coral rubble, shell hash, cobbles, gravel and/or sand are widely distributed across the globe. They have developed along marine and lakeshores under favourable wind and wave conditions, and sufficient long-term sediment supply.

During the past few decades, research on beach-ridge sequences has progressed from describing their morphology and possible origins (Taylor and Stone, 1996; Otvos, 2000) to enabling their usage for palaeoenvironmental reconstructions. They can be used to assess external controls of (relative) sea-level rise, land subsidence, variations in storm impact, and changes in climate and upstream land use (Scheffers et al., 2012; Tamura, 2012 and references therein). They also may include markers left by catastrophic events like volcanic eruptions (Nieuwenhuyse and Kroonenberg, 1994; Nooren et al., 2017), and host soils that are suitable for chronosequence studies (Nielsen et al., 2010; May et al., 2015; Hinojosa et al., 2016).

The number of preserved ridges determines the extent of the palaeo-environmental record stored in the associated sediments, with resolutions up to decadal scale (cf. Curray et al., 1969; Nielsen et al., 2006; Milana et al., in press). The largest beach-ridge plains with multiple parallel beach ridges are formed along medium- to low-energy shorelines of lakes and seas. The beach-ridge plain on the seaward margin of the terrestrial Usumacinta-Grijalva delta in southern Mexico (Fig. 1a) is probably the world's largest. Since the strong reduction in the rate of postglacial sea-level rise in the mid-Holocene, hundreds of semi-parallel sandy beach ridges formed across a shore-perpendicular distance of more than 20 km. In our study area near Frontera (Fig. 1b) beach ridges include aeolian topsets composed of backshore-fringing foredunes. In this paper, we use Otvos's (2000) broad definition of beach ridges, including all 'relict, semi-parallel, multiple ridges' formed by waves (berm ridges), wind (multiple ridges originating as foredunes) or a combination of both.

Earlier morphological studies (Psuty, 1965, 1967; West et al., 1969) identified three main phases in the development of the beach-ridge plain, each linked to a specific position of the rivers' main channels (Fig. 1b). The north-easterly branches of the Grijalva fan-delta river system created favourable conditions for local beach-ridge-complex initiation and development during Phase 1, the Usumacinta (with the San Pablo y San Pedro River (SP y SP in Fig. 1b as the main outlet) during Phase 2 and both rivers (though a combined outlet near Frontera) during Phase 3. Psuty (1965, 1967) proposed an important role to storm surges and overwash in the formation of the beach ridges. Aguayo et al. (1999) established a preliminary chronology of beach-ridge formation on the basis of radiocarbon-dated bivalves and gastropods. Our study elaborates on these pioneering works, aiming to establish a robust chronology for the beach-ridge sequence and to understand the apparent periodical variations in beach-ridge height that are seen in LiDAR imagery of the study area (Fig. 2a).

In the long-term ($10^3$ years), the considerable accretion of the beach ridge complex has been driven by steady sediment supply by the Usumacinta and Grijalva Rivers (West et al., 1969). Much of this sediment has been generated in their upper catchments and routed through the delta plain to the coastal zone. Morphometric variations between the main phases of beach-ridge formation (Fig. 1b) is mainly influenced by spatiotemporal variability in the positions of the river mouths, size of the feeding river and magnitude of sediment fluxes carried by the water. Studies on other beach-ridge systems suggest that shorter term ($10^1$-$10^2$ years) variability can reflect oscillations in river-mouth sediment supply (Brooke et al., 2008a; Tamura, 2012), potentially



making the Usumacinta-Grijalva beach-ridge sequence a proxy record for variability in
precipitation in the hinterland.
To test this hypothesis, we conducted a detailed geomorphological and sedimentological field
study, linking LiDAR data to cored and geophysically surveyed transects, and extensive
sediment analyses and dating. Our study covers 150 km of the beach-ridge complex in a shore-
parallel direction and 20 km in a shore-normal direction. Grain-size and mineralogical analyses
are potentially powerful tools to understand transport and deposition mechanisms of beach-ridge
sands (cf. Visher, 1969), but have scarcely been applied in recent beach-ridge studies (exceptions
are Guedes et al., 2011; Garrison et al., 2012). Volcanic glass and pumice fragments are highly
informative components of the beach-ridge sands (Nooren et al., 2017), and have been
chemically fingerprinted to determine their provenance. The internal architecture of the beach
ridges was imaged with ground-penetrating radar (GPR), as in other beach-ridge and coastal-
barrier studies (e.g. Jol et al., 1996; Van Heteren, 1998; Bristow and Pucillo, 2006; Forrest,
2007; Oliver, 2016).
A detailed chronology of the sequence was established from the combined deployment of
Optically Stimulated Luminescence (OSL) on quartz grains (quartz content of the sand is 50 to
65%, Aguayo et al., 1999), and AMS $^{14}$C dating of thin layers of terrestrial organic debris (leaf
fragments) in the beach-ridge sand. Here we expand on the chronology of a 3-km-long beach-
ridge subsection documented in Nooren et al. (2017). Quartz-grain OSL dating has been widely
used for establishing the age of coastal deposits in general (e.g. Ballarini et al., 2003; Nielsen et
al., 2006; Reimann et al., 2011) and beach-ridge sequences in particular (Tamura, 2012 and
references therein; Oliver et al., 2015; Rémillard et al., 2015; Vespremeanu-Stroe et al., 2016;
Milana et al., in press), but its combination with AMS $^{14}$C dating of thin organic debris layers is
presented here for the first time. It provides a unique opportunity for cross-validating the
methods.
**2 Geographical Setting**
The study area is part of the beach-ridge system along the edge of the Holocene Usumacinta-
Grijalva delta plain, and stretches from Paraiso in the west to Ciudad del Carmen in the east (Fig.
1b). The delta plain and its hinterland have a humid tropical climate with mean annual
precipitation ranging from 1000 to 1500 mm in the highlands of the Chiapas Massif and along
the Tabasco coast to locally more than 5000 mm in the mountain foothills in between (West et
al., 1969; Hijmans et al., 2005). Approximately 80 % of the annual precipitation falls in a rainy
season that lasts from June until November. The excess or effective precipitation contributing to
river discharge is around 40-60 % (Table 1). Peak discharges are related to the passage of large
tropical depressions, most frequently occurring in September and October.
The drainage basin of the Usumacinta River is dominated by a Cretaceous limestone plateau,
folded during the Paleogene (Padilla and Sanchez, 2007), with elevations rarely exceeding 700 m
above mean sea level (m+MSL). The headwater catchments of this river, however, are composed
of pre-Mesozoic plutonic, metamorphic and volcanic rocks (Fig. 1a). These uplands are dotted
with large remnants of Los Chocoyos ignimbrites left by a Pleistocene caldera-forming eruption
at Atitlán volcanic centre in southern Guatemala. The Los Chocoyos ignimbrites are also found
in the upper drainage basin of the Grijalva River, up to 130 km from the Atitlán caldera
(Sánchez-Núñez et al., 2015), but they do not have the same extent as the deposits within the
Usumacinta drainage basin.



Presently, routing of sediment from upstream to downstream reaches of the Usumacinta River is
blocked by the Chixoy hydroelectric dam at Pueblo Viejo (Fig. 1a). This man-made obstacle has
reduced sediment transport to the coast since its completion in 1983. High erosion rates have
caused rapid infill of the reservoir behind the dam. Between 1983 and 2009, approximately
$158 \cdot 10^6$ m$^3$ of sediment has accumulated at an average rate of $6.1 \cdot 10^6$ m$^3$/year (Jom Morán,
2010). The total volume of upland source material and the rate at which it is transported
downriver show that the Usumacinta could have contributed a sufficient amount of sediment for
the rapid progradation of the beach-ridge plain. Nieuwenhuyse and Kroonenberg (1994)
demonstrated a similar important role of volcaniclastic sediments in the formation of Holocene
beach ridges in Costa Rica.
The coastal zone experiences a diurnal tide with a microtidal range between 0.25 and 0.75 m.
During most of the year, low-energy waves coming from the northeast with swells of 0.3 to 0.7
m produce a wave-generated longshore current carrying river sediments westwards (West et al.,
1969). Under these fair-weather conditions, beach accretion is common (Psuty, 1965, 1967),
building out the promontories of active river mouths. Usually some 20 to 25 'Nortes' or frontal
storms hit the area between October and March. These produce strong north-westerly winds
generating swells of 1.2 to 1.7 m as well as local longshore-current reversals and commensurate
beach erosion (West et al., 1969). Wave climate increases westward in the dominant longshore-
current direction, a result of relatively steeper shoreface slopes in the western part of the study
area (notice 10-m depth contour in Fig. 1b). Newly formed beach ridges are rapidly colonised
and stabilised by vegetation, most noticeably and dominantly by *Ipomoea pes-caprae*, a salt-
tolerant coastal pioneer species (Castillo et al., 1991; Gallego-Fernández and Martínez, 2011).
## 3 Materials and Methods
### 3.1 Geomorphological and sedimentological survey
The LiDAR data (Fig. 2a) were originally acquired in April-May 2008 and processed by
Mexico's National Institute of Statistics and Geography (INEGI). The derived DEM product has
a cell size of 5x5 m, has cm-scale vertical resolution and is accurate to 0.15-0.30 m (Ramos et
al., 2009). The LiDAR imagery is used to morphometrically distinguish main and sub-phases of
progradational beach-ridge formation, focusing on internal similarity in ridge dimensions,
orientation, and lateral and cross-cutting relationships with river-channel morphology. We
identified and defined sub-phases that correspond to periods of relatively stable river-mouth
configurations, with smaller and larger river-network reconfigurations as the breaks between.
Avulsions affecting the main river branches have drastically changed their position in several
instances, and, consequently, the supply of sediment to the beach-ridge system. This changing
supply is particularly recognisable from the truncation of beach ridges of former river
promontories at the modern coastline, but can also be seen from orientation shifts in beach-ridge
alignments within the beach-ridge complex.
LiDAR-inferred morphometric phases were ground-truthed using sediment composition and
chronometric results from four field campaigns in the period 2011-2015. To describe and sample
the sandy, waterlogged lithology, sediment cores reaching 4 to 11 m depth were taken with a soil
auger and a Van der Staay suction corer (Van de Meene et al., 1979). Boreholes were placed
along three shore-normal (A, B and C) and two shore-parallel (D1 (youngest beach ridge) and
D2) transects (Fig. 2a). To support the interpretation of the grain-size data, surficial nearshore



sediments were sampled off Playa La Estrella in April 2013 for modern-analogue study of the
shore-normal sorting processes.
The shore-parallel transects aimed at characterising the aeolian facies encountered on the most
recent beach ridge, and the swash facies encountered at ~1 m below MSL in a relatively elevated
fossil beach ridge. The shore-normal transects aimed at establishing the progradational
chronology and its relation with river shifts, with densest sampling along Transects A and B
(Fig. 2b). A 3-km-long subsection of Transect A, containing evidence for a volcanic eruption of
El Chichón in 540 CE, was studied in substantial detail (Nooren et al., 2017). For consistency,
each coring location was chosen at the seaward foot of an individual ridge, except when the
aeolian cap on top of the ridges was sampled. Bagged samples of sand were collected at 0.2-0.5
m core-intervals. Encountered organic debris-rich layers were sampled and stored in a cold room
($4^oC$) pending further processing for AMS $^{14}C$ dating. For OSL dating, nineteen samples were
collected in 30-cm-long opaque tubes from the bottom of shallow hand-augered boreholes during
the dry seasons of 2012 and 2013. OSL sample 450 was collected from a soil pit dug in a beach
ridge for use in a chronosequence study (Hinojosa et al., 2016).
More than one thousand sand samples were collected in the field, transported to the Netherlands,
dried at 105 $^oC$, and stored at room temperature. Magnetic susceptibility was measured on all
dried sand samples with a hand-held ZH Instruments SM 30. Calcium carbonate was measured
on sand samples from the two shore-parallel transects and on sand samples from cores 192, 252,
432, 433, 435, 452 and 453 (Fig. 3), to estimate the maximum depths of pedogenic
decalcification, which indicates the position of the phreatic surface (ground water level and, by
proxy MSL). Calcium carbonate was measured with a Scheibler Calcimeter, by adding 10% HCl
solution to 1 g sediment and measuring the produced $CO_2$ volumetrically. Carbonate content is
expressed as weight percentage $CaCO_3$. Grain-size analyses (range 0.15 – 2000 μm) were
conducted with a Sympatec HELOS/KR laser diffraction particle sizer, equipped with an
advanced wet disperser (QIXEL). Before measurements, organic matter and carbonates were
removed with 20% $H_2O_2$ and 10% HCl. Grain-size parameters (median, sorting, skewness and
kurtosis) were calculated following Folk and Ward (1957).
Grain-size and magnetic-susceptibility investigations were supported by a limited number of
heavy-mineral analyses to characterise the source material. Heavy minerals were separated with
a heavy-liquid solution (Sodium Polytungstate, $Na_6[H_2W_{12}O_{40}]$) with a density of 2.85g/cm$^3$, and
identified under a polarised-light microscope. Volcanic glass shards and a pumice clast retrieved
from four beach-ridge cores along Transect A, covering a large temporal range in beach-ridge
formation (Fig. 2b and 3a, samples 336, 252, 193 and 197), were chemically fingerprinted to
identify the eruption source(s). Major-element compositions of the glass shards were determined
on 5-12 particles per sample with a Jeol JXA 8600 microprobe equipped with five wavelength-
dispersive spectrometers. Measurements were performed by WDS using 15kV acceleration
voltage, 10nA beam current and a defocused beam (5μm spot size) to minimise mobilisation of
sodium. Instrumental performance and calibration were monitored by repeated analyses of
natural glass standards (rhyolitic USNM 72854 VG-568 and basaltic USNM 111240 VG-2) and
in-house mineral standards.
**3.2 AMS radiocarbon and OSL dating**
Within the beach ridges, 1- to 5-cm-thick layers of organic debris were commonly found,
especially at locations relatively close to a (former) river mouth (Transects A and B3). The layers
contained charcoal, wood and leaf fragments, often mixed with shell fragments. This organic
material is transported to the coast by the rivers, then further distributed by longshore currents to



eventually be incorporated into the beach ridge facies. The debris is a mixture of apparently
younger (hardly harmed) and older (rounded edges) reworked material. Reworking was
especially evident from the commonly rounded edges of wood and charcoal fragments in the
detritus cocktail. Reworked organic material was purposely avoided in our sampling (apart from
test samples to demonstrate the associated danger of age overestimation) and age-distance
modelling.

Thirty-five terrestrial macro-remains (mainly leaf fragments), isolated from organic debris
layers, were standard AAA pretreated, and $^{14}$C dated using an AMS facility (Van der Plicht et al.,
2000). Ages were reported in yr BP, using the Libby half-life and corrected for isotopic
fractionation via $\delta^{13}$C (Mook and Van der Plicht, 1999). They were calibrated with the software
package OxCal 4.2 (Bronk Ramsey, 2009) using the IntCal13 calibration curve (Reimer et al.,
262  2013).


Twenty OSL samples were dated using Risø TL/OSL DA15/20 readers (Bøtter-Jensen et al.,
2003), equipped with Sr/Y beta sources. About 130 g material from the (light-exposed) outer
parts of the sample tubes was used for dose-rate determination. High-resolution gamma
spectrometry was used to determine radionuclide-activity concentrations ($^{40}$K, and several
nuclides from the U and Th decay chains). Measured values were converted to environmental
dose rates using conversion factors of Guerin et al. (2011), assuming immediate burial of the
samples to present depth, and accounting for attenuation due to water and organic material
(Aitken, 1998) and cosmic-ray contributions (Prescott and Hutton, 1994). For OSL samples
obtained from below the groundwater table, a water content of $25 \pm 5\%$ by weight was used
(pore space fully water saturated), assuming permanent saturation over the entire burial period.
For some of the older samples, it is likely that they were deposited above contemporary
groundwater levels (Fig. 3b). However, at this stage it is not possible to make a more realistic
estimation of the average water content over the entire burial period. Dependency of dose rates
and hence OSL ages on water content, implies that OSL age estimates will decrease by
approximately 1 % for each weight % decrease in water content (Aitken, 1998). For two OSL
samples taken above the groundwater table, a water content of $5 \pm 3 \%$ was used (moisture
contents at field capacity).

OSL samples were prepared following standard procedures including sieving and chemical
treatment with $H_2O_2$, HCl and HF, to yield sand-sized purified quartz of 212–250 μm. For
aeolian sample 179, the fraction 180-212 μm was used. Quartz OSL signals were detected
through a 7.5 mm Hoya U340 filter, and an early background approach was applied to obtain a
net signal that is dominated by the fast OSL component of quartz (Cunningham and Wallinga,
2010). The OSL IR depletion ratio of Duller (2003) was used to check for feldspar
contamination. Equivalent doses were determined on small aliquots (2 mm, ~60 grains) using the
Single Aliquot Regenerative dose procedure (Murray and Wintle, 2003). The Central Age Model
(CAM, Galbraith et al. 1999) was used to determine over-dispersion in the resulting equivalent-
dose distributions (i.e. spread in results on individual aliquots that is not explained by the
analytical uncertainties) and for burial-dose estimation. In case of high over-dispersion (>30%)
in combination with skewed dose distribution, the burial dose was estimated using a
bootstrapped version of the Minimum Age Model (Cunningham and Wallinga, 2012). OSL ages
are determined by dividing the sample burial dose by the sample dose rate and reported in Year
CE, with 1-sigma uncertainty ranges. For each sample, validity of the OSL age was assessed on
the basis of the equivalent-dose distribution.

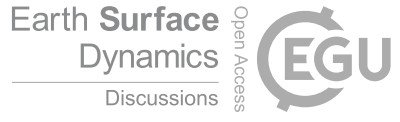

The full set of calibrated AMS [14]C and OSL ages was used to establish an age-distance model,
using the P_sequence module of the Oxcal 4.2 programme (Bronk Ramsey, 2009; 2016). We
furthermore demonstrate the variability in age-distance models for part of Transect B if we
assume a constant aeolian accretion rate, following the approach of Minderhoud et al. (2016).
**3.3 Ground-penetrating radar**
GPR surveys were conducted at the end of the dry season in June 2012 along parts of the
transects (Fig. 2a). Data were collected using a MALA ProEx system with 250-MHz shielded
antennas and an odometer wheel for accurate positioning (0.1 m step size). For the time-depth
conversion, we used signal velocities of 0.125 (based on the move-out of diffraction hyperbolas)
and 0.06 m/ns for deposits above and below the groundwater table, respectively.
**3.4 Beach-ridge elevation and accretion volumes**
Fifteen cross-normal ribbon-shaped elevation transects (Fig. 2b) were sampled from the LiDAR
based DEM, and combined with the dating information to calculate the temporal variability in
beach-ridge elevation and accretion volumes. To exclude short-term variability in beach-ridge
elevation and to minimise the effect of local erroneous elevation values we divided the 1-km-
wide ribbons into multiple polygons (Fig. 2b). Each polygon included at least one, but on
average a few ridge/swale couplets.
We estimated an average thickness for the Holocene beach-ridge deposits of 10±2 m, based on
geophysical tests conducted near the current combined Usumacinta-Grijalva River outlet
(Administración Portuaria Integral de Dos Bocas S.A. de C.V., 2005). Unfortunately, we have
limited information regarding the inland spatial variability in thickness of the beach ridge
complex, and our deepest Van der Staay core of 11 m (core 426, Figs. 3a and 4) did not penetrate
the base of the Holocene beach-ridge deposits at this location.
Aeolian accretion sub-volumes were calculated from the ribbon-averaged estimated mean beach-
ridge elevation. The calculation assumed all sandy deposits above an estimated average swash
run-up height of 0.5 m above MSL at the time of beach-ridge formation to be aeolian in origin.
We used our decalcification depth observations (which sits decimetres deeper than the current
groundwater level at more inland beach ridges) and the resemblance of this signal with Gischler
and Hudson's (2004) sea-level curve for Belize, to assess the MSL positions at the time of
beach-ridge formation. The calculations were performed for Phase 2 and Phase 3. Along
Transect A we added 1 m to the raw LiDAR DEM values because the surface elevations as
estimated during the fieldwork period were systematically 1 m higher than the first-generation
DEM product for this subarea. We assume that the groundwater level by the end of the dry
season in 2012 and 2013 should at least correspond to or be above present MSL, as was the case
at core locations along Transects B and C.
**4 Results**
**4.1 LiDAR DEM analyses**
The three main phases in beach-ridge formation (Psuty, 1965, 1967; West et al., 1969) are easily
discernible from the LiDAR-based DEM (Fig. 2a). Approximately 500 beach ridges can be
distinguished. Their spacing is typically between 20-100 m, and mean surface elevations along
the three shore-normal transects vary between 0.5 and 3.5 m+MSL (Fig. 3). Beach ridges are
relative low and widely spaced near (former) river mouths. Away from a river mouth they merge
or become more closely spaced. Beach-ridge elevation, however, tends to increase with distance



from a river mouth. The most elevated beach ridges (up to 5 m+MSL) are found in the western
part of the study area (Fig. 2a) – on the downdrift side of the system. The influence of drift
direction is also apparent in the modest asymmetry of the truncated Phase 2 promontory at the
mouth of the SP y SP River and in the strong westward deflection of the mouth of the Gonzalez
River (Fig. 2b).
Two faults (Fig. 2b), almost perpendicular to the orientation of the beach ridges, may be
responsible for the slight east-west tilt of ridges in this part of the study area. The DEM shows no
evidence for any significant horizontal displacement along NW-SE oriented strike-slip faults
described by Aguayo et al. (1999).
Scour holes, possible features produced by large storm surges, are clearly identifiable along only
one beach ridge in the western part of the study area (Fig. 2b), and washovers are not apparent
from the DEM, indicating that few extreme storm events left clear traces in the area.

**4.2 Beach-ridge chronology**

The 35 AMS $^{14}$C and 20 OSL sample ages (Figs. 2b, 3 and 4, Tables A1 and A2) offer a
significant refinement of the preliminary beach-ridge chronology proposed by Aguayo et al.
(1999) on the basis of radiocarbon-dated shell material. The resolution offered by the large
number of dated samples facilitated the development of age-distance models for the progradation
of the beach-ridge plain (Figs. 3 and 4), used in turn to reconstruct the palaeoshorelines as
indicated in Figure 5a.

The sequence of calibrated $^{14}$C ages shows very good internal consistency, with only two
statistically significant age reversals (both in Transect A2; Fig. 4c). This more than fair
agreement of $^{14}$C ages with vertical stratigraphic order and lateral geographic position gives
confidence to their representativeness for deposition age. Nevertheless, dated organic detrital
fragments give 'Terminus Ante Quem' ages that may be older than the beach-ridge sand in which
they were entrained. Charcoal fragments have been found to be many hundreds of years older
than the more fragile leaf fragments from the same debris layer (Fig. 3a and Table A1, sample
252 and 336), and do not provide a reliable age of final deposition. We therefore avoided wood
and charcoal in our sample analysis and only used dated leaf fragments for the age-distance
models (Fig. 3). Of all the terrestrial macro-remains in the organic debris layers, fragile leaves
are assumed to be the least likely to have survived repeated reworking. There are some
indications, however, that even the leaf fragments have undergone some reworking, because
samples taken farther from the former river mouth in Transect B2, appear to be 200-500 years
older than the LiDAR-tracing projected AMS $^{14}$C ages of samples taken closer to the river mouth
in Transect A (Figs. 3b and B1).

Quartz OSL behaviour of the samples showed suitability for dating. A dose-recovery experiment
indicated that a given dose could be retrieved accurately (dose-recovery ratio $0.997 \pm 0.014$,
n=39). Equivalent-dose distributions were normally distributed and showed over-dispersion as
expected for well-bleached deposits (average 18%, n=17). For three samples (179, 427 and 444),
higher over-dispersion (>30 %) was observed. The reliability of samples 179 and 427 was
considered questionable because the equivalent-dose distributions lacked the characteristic
skewness that would characterise over-dispersion due to heterogeneous bleaching (e.g. Wallinga,
2002). For sample 444, heterogeneous bleaching was inferred from the large over-dispersion in
combination with positive skewness in the equivalent-dose distribution. For this sample a burial
dose was determined using a bootstrapped version of the Minimum Age Model (Cunningham
and Wallinga, 2012), resulting in a higher-confidence OSL age.



Dose rates were found to vary between 1.83 ± 0.08 and 2.66 ± 0.10 Gy/ka (mean 2.18 Gy/ka).
These values are lower than those reported for Usumacinta levee deposits (2.38 – 4.55 Gy/ka,
Muñoz-Salinas et al., 2016). The difference is likely related to lower amounts of silt and clay in
the beach ridges than in the levees. Dose rates are much higher than the extremely low values
reported for the quartz-rich beach ridges in Florida (e.g. Otvos, 2005; López and Rink, 2008;
Rink and López, 2010).

Quartz OSL ages are internally highly consistent, and agree well with the calibrated [14]C ages
(Figs. 3 and 4), underscoring the usefulness of OSL dating in the establishment of beach-ridge
chronologies (cf. Tamura, 2012).

For two samples (451 and 450), collected at the same location but at different depths, OSL
results suggested an age difference of about 600 years. A possible partial explanation is that the
water-content estimations for these samples (field capacity for OSL sample 451; water-saturated
for sample 450) (Table A2) are not correct. If more similar water contents are assumed for both
samples, the age difference is much reduced, highlighting the importance of water-content
estimation in OSL dating. An alternative, or additional, explanation could be that the sediment
above the groundwater table was reworked (e.g. through bioturbation). The spread in equivalent-
dose distribution for sample 179 may indicate such reworking, but for sample 451 the
equivalent-dose distribution provides no evidence of reworking. For the age-distance model, we
excluded OSL ages that were judged to be of questionable validity (179 and 427) and those
obtained from sediments above the groundwater table (179 and 451).

The age-distance models for Transects A and B are presented in Fig. 3. For a 3-km section
(Transect A2), the age-distance model was published by Nooren et al. (2017; Fig. 4c). Three new
OSL analyses (this paper; Table A2 and Fig. 4c), one providing a questionable age (sample 427),
corroborate the robustness of that study. Radiocarbon ages of shells reported by Aguayo et al.
(1999) do not provide additional age constraints, owing to limitations in accuracy of the shell
ages caused by carbon reservoir effects and taphonomic depositional uncertainty.

We ran a P_sequence Bayesian calibration model (k=0.05 m[-1]) (Bronk Ramsey, 2009), fed with
the AMS [14]C and OSL dates and relative shore-normal positions, and with boundaries (i.e.
discontinuities) prescribed at the transitions between the three main beach-ridge-formation
phases. For the age-distance model of Transect B (Fig. 3b), we projected AMS [14]C and OSL ages
of samples from Transect A, correlating along the beach-ridge traces in the LiDAR data. Because
of the assumed time lag between the final burial of leaf fragments in the beach ridges at smaller
(Transect A) and greater (Transect B) distance to the river mouth during Phase 2, in the
corresponding part of Transect B the [14]C ages of samples 185 and 438 (Fig. 3b) were excluded
from the model. We identified one OSL age (sample 437) as an outlier (too old compared to ages
of neighbouring samples) and excluded it from the age-distance modelling (Fig. 3b).

The age-distance model for Transect A (Fig. 3a) shows a long-term average progradation rate
that decreased from 4.1 to 3.4 m/y between the start of Phase 2 (~1800 BCE) until the transition
between Phases 3A and 3B, dated at ~1050 CE. Progradation rates returned to higher values
during Phases 3B and 3C, 4.0 and 4.5 m/y respectively, related to the reconfiguration of the river
system and the avulsion of the Usumacinta River around 1050 CE (discussed in section 5.1).

The age-distance model for Transect B (Fig. 3b) includes a preliminary model for Phase 1 (4500
–1800 BCE). The model is based on relatively few samples, including OSL ages sensitive to
uncertainty related to water-content assumptions, and must therefore be treated with caution. The



age-distance model for Phase 2 has an age range between 1775 ± 95 BCE and 30 ± 95 CE (at
1σ), which covers a slightly shorter time period than at Transect A where Phase 2 runs until
approximately 150 CE. The LiDAR image shows clear signs of truncated beach ridges between
Phases 2 and 3 at Transect B, explaining the occurrence of a hiatus. To investigate possible age-
distance scenarios for Transect B (Phase 2), we calculated five possible short- and long-range
scenarios (Transect B2-1 till B2-5 in Fig. 2b) by including aeolian accretion (see section 4.6) as a
proxy for progradation rate of the beach-ridge plain. The depicted scenarios (Appendix B, Fig.
B1) assume shore-normal aeolian accretion activity to be constant between 1800 BCE and 30
CE. Under this assumption the most noticeable change in progradation rate occurred around
1000 BCE, during a period when relatively high beach ridges are indicative for a strong drop in
progradation rate. This is apparent in both long- and short-range scenarios and at all five
transects. The long-range scenarios seem to be in better agreement with the mean of the OSL
ages. These calculations show the potential to improve age-distance models with additional
information regarding the temporal variability in aeolian accretion rates.
The age-distance model is less reliable for Phase 3A owing to the lack of dated samples along
Transect B, the rejection of OSL sample 179 and uncertainties in the projected location of dated
samples from Transect A. The age-distance model is very robust again for the period 1050 CE to
present (Phases 3B and 3C), with precision of modelled ages in the order of only 10–60 years (at
1σ).
For Transect C the age-distance model (not shown) is preliminary, because it only relies on two
AMS [14]C dated samples (Table A1), and geomorphological age-projections from Transect A.
**4.3 Grain-size analyses**
The beach ridges consist of moderately well- to well-sorted fine to medium sand. All samples
show a unimodal grain-size distribution with a median between 117 and 350 μm (Fig. 5b). The
grain-size of sand samples from two shore-parallel transects (Fig. 6) show a general coarsening
in the dominant (westward) longshore-transport direction.
The longshore trend in grain size is apparent in both swash and aeolian facies (Fig. 6), applies
along the full length of the study area, and does not appear to be affected by the deltaic
promontory of the Usumacinta/Grijalva River in the middle of it. Skewness of the grain-size
distribution increases in the dominant longshore-transport direction, denoting an increase in
excess fines, and the swash facies tends to get better sorted (decrease in phi values) in the same
westward direction. Kurtosis values do not show systematic changes. Magnetic-susceptibility
values also tend to increase in a westward direction, with the most elevated values around the
(former) waterline, as heavy minerals, including titanomagnetite, preferentially accumulate in the
swash zone (Komar, 2007). The high magnetic-susceptibility values for aeolian beach-ridge sand
near the mouth of the currently active Usumacinta/Grijalva and Gonzalez Rivers is likely related
to the contribution of volcaniclastic material from El Chichón's 1982 eruption, as magnetite
enrichment in the beach-ridge sands also occurred after earlier eruptions of El Chichón (Nooren
et al., 2017). The $CaCO_3$ concentration decreases in the longshore transport direction, in line
with a decreased influence of calcareous sediment from the calcareous platform in the eastern
part of the study area (Ayala-Castanares and Guittiérrez-Estrada, 1990) (Fig. 1b).
The westward increase in median grain size probably relates to an increase in wave energy,
which also may have caused the steepening of the shoreface slopes in that same direction. The
presence of mega-cusps at beaches near the mouth of the Gonzalez River is an additional
indication of relatively strong wave impact on the western side of the system. Similarly, and at



first sight contradictory, grain-size coarsening in the longshore-drift direction was observed at
Sint George Island (Balsillie, 1995) and along the North Sea beaches of East Anglia, England
(McCave, 1978). McCave (1978) explained the coarsening of beach sand in the longshore-
transport direction as a result of the winnowing of fines and their offshore transport by tidal
currents. Similar processes could be responsible for the westward grain-size coarsening, and
could explain the dominance of relatively fine clastic sediments on the continental shelf at the
study site (Ayala-Castanares and Guittiérrez-Estrada, 1990) (Fig. 1b). The offshore transport of
fines is probably stimulated by the anticyclonic eddy that develops during spring and moves
westward along the coast during summer (Salas de León et al., 2008). This eddy influences
bottom currents, especially west of Usumacinta/Grijalva River outlet. Lastly, it should be noted
that deviations from this general pattern in longshore grain-size distribution do occur. The
relatively coarser grain size of the three aeolian samples approximately 10 km west of the SP y
SP River for example are probably due to the contribution of eroded and reworked sand from the
old promontory of the SP y SP River (Fig. 6).
Although the major variability in grain-size parameters occurs in a shore-parallel direction,
shore-normal sorting processes due to wind and wave activity have resulted in significant
variation in grain-size parameters as well (Fig. 7). Surface samples from the modern beach
profile at Playa Estrella (Fig. 7a) show an increase in grain size from offshore towards the coast,
with coarsest and least-sorted sand occurring in the relatively high-energy swash zone. The
grain-size characteristics of backshore beach deposits and dune/ridge sands are very similar.
They differ from the swash deposits in having a reduced presence of coarse grains and a better
sorting (Fig. 7). These properties indicate that aeolian processes likely have been in play in the
development of backshore deposits and dune ridges.
The grain-size variability in shore-normal direction along Transect A (Appendix B, Fig. B2) is
very similar to that of surficial samples taken at the current beach at Playa Estrella. Samples
from core 197 (Fig. B2, 0.04 km) reflect shore-normal sorting processes and demonstrate a
coarsening-upward sequence with strongly negatively skewed relatively fine sandy deposits at –4
m+MSL, likely deposited in the nearshore zone (Fig. 7a). These deposits are covered by a few
meters of fine sand with grain-size parameters resembling the surficial samples from the swash
zone (Fig. 7a), consistent with Walther's Law.
Samples from beach ridges formed during Phase 3B (Figs. 4b and B2, 3.5 km) are strikingly
different from the general pattern (Fig. 7b), with a higher contribution of well-sorted fine to
medium sand, likely related to an increased availability of reworked sand due to the erosion of
the SP y SP promontory. The same process is likely responsible for the coarser grain sizes of the
aeolian sand samples from the youngest ridge collected 10 km west of the still eroding SP y SP
promontory (Fig. 6).
### 538 4.4 Internal architecture

Despite the high signal attenuation, which limited the depth of investigation in various areas, the
GPR measurements clearly show strong seaward-dipping reflectors in all transects (Fig. 8), with
slopes between 2 and 5° (Fig. 4b and 8). Since all GPR transects were oriented perpendicular to
the ridges, these angles are close to the actual angles. The values are similar to dipping angles
reported by Psuty (1967) for beach deposits elsewhere along this coast. The largest slope angles
are preferentially associated with more elevated beach ridges. No reflections hinting at
interrupting erosional surfaces are apparent, and strong landward-dipping reflectors were rarely
encountered in the GPR-surveyed transects.



The top of the foreshore deposits is located around 0.8 m+MSL (Fig. 8). At depths between 1
and 2 m-MSL, the slopes of the upper-shoreface deposits start to decrease. Reflection
terminations (e.g. at x = 40 m and y = 60-80 ns; x = 85 m and y = 35 ns in Fig. 8) suggest the
periodic welding of bars onto the beach face (i.e. beach progradation by bar accretion). The few
landward-dipping reflections seen at the top of the beach sequence presumably relate to the infill
of a large runnel that formed when a swash bar merged with the beach.
The GPR results compare well with the extensive investigations conducted at the fine sandy
swash-built beach ridges at St. Vincent Island, Florida (Forrest, 2007), confirming the
prominence of swash deposits in beach-ridge sequences formed under microtidal conditions and
relatively low wave impact. It is hard to distinguish the aeolian radar facies from that of the
lithologically similar beach deposits, with the only useful indicator being the termination of
seaward-dipping foreshore reflections (red dashed line in Fig. 8). The absence of significant
internal erosional surfaces suggests that the ridges formed quickly or at least continuously,
uninterrupted by significant coastal-erosion events. Landward-dipping overwash deposits, as
described by Psuty (1967; 1969), are not evident in our selected GPR transects (nor did LiDAR
data support their presence in the promontory parts of the beach-ridge complexes). The
landward-dipping structures in Fig. 8 are situated too deep in the subsurface to be interpreted as
overwash deposits.

**4.5 Composition and source of beach-ridge sands**

The major-element compositions of relatively large sand-sized volcanic glass shards and pumice
fragments (250-1500 μm) and a pumice clast of 1.5 cm, isolated from beach-ridge samples along
Transect A, are reported in Table A3. The major-element composition is similar to that of the
Late Pleistocene Los Chocoyos tephra (Kutterolf et al., 2008), and is significantly different from
any of the late-Holocene tephras of El Chichón volcano (Fig. 9) (Nooren et al., 2017). It is
therefore inferred that Los Chocoyos ignimbrites have been an important sediment source for the
Usumacinta-Grijalva delta. They were emplaced during a mega-eruption at Atitlán volcanic
centre around 84,000 years ago (Drexler et al., 1980), which produced an estimated 150 to 160
km$^3$ Dense-Rock Equivalent (DRE) of tephra fall and some 120 km$^3$ DRE of pyroclastic flow
deposits (Rose et al., 1987). It is the only Late-Pleistocene volcanic eruption that deposited
voluminous tephra north of the Motagua River valley (Fig. 1a; Koch and McLean, 1975). The
Los Chocoyos pyroclastic flow deposits reach thicknesses of more than 200 m, and have been
found well into the watersheds of the Grijalva and Usumacinta Rivers (Instituto Geográfico
Nacional, 1970; Koch and McLean, 1975; Rose et al., 1987; Sánchez-Núñez et al., 2015). We
estimate that approximately 3 % and 16 % of the pyroclastic flow deposits were deposited in the
Grijalva and Usumacinta watersheds, respectively. In the steep and poorly vegetated terrain,
these volcaniclastic deposits are vulnerable to erosion and particularly prone to mass transport by
landslides (Harp et al., 1981). It is therefore not surprising that abundant volcaniclastic minerals
and glass shards (Solis-Castillo et al., 2013) were found in Holocene levee deposits of the
Usumacinta River at Tierra Blanca (Fig. 1a), reflecting reworked Los Chocoyos tephra, as
geochemical and micromorphological evidence suggests (Table A3, (Cabadas-Báez et al., in
press).
The heavy-mineral analyses confirm the presence of volcaniclastic material within the beach-
ridge sands. The non-opaque heavy minerals are dominated by green and brown amphiboles,
clinopyroxene, titanite and epidote, whereas the opaque heavy minerals are dominated by
titanomagnetite.





### 4.6 Beach-ridge elevation

The temporal variability in beach-ridge elevation along the fifteen cross-normal ribbon-shaped elevation transects representing Phases 2 and 3 is demonstrated in Fig. 10. Most noticeable are the high-amplitude elevation changes along Transect B during Phase 2, and the relatively low standard deviations during periods in which elevated beach ridges were formed. Overall, mean swale elevations along Transects A, B and C show a continuously increasing trend of about 0.3 mm/yr (Fig. 10), which is in line with expected long-term rate of relative sea-level (RSL) rise in the southern Gulf of Mexico area, and comparable to those of the reconstruction of RSL rise made by Gischler and Hudson (2004) for Belize. The estimated depths of pedogenic decalcification (Figs. 3a and 3b) also supports this RSL curve, but further analyses are needed for better refinement. We found no evidence for a mid-Holocene RSL high-stand followed by a 2-m drop during the late Holocene (e.g. Stapor et al., 1991; Tanner, 1992; Morton et al., 2000; Blum et al., 2003). Rather, our observations are in accord with more recent RSL reconstructions for the northern Gulf of Mexico coast that show a gradual rate of RSL rise during the late Holocene (Törnqvist et al., 2004; Milliken et al., 2008; Donelly and Giosan, 2008).

### 4.7 Volumetric growth rate of the beach-ridge plain

The total average late-Holocene sediment-accumulation rate was estimated by simply dividing the total volume of beach-ridge deposits along the system's 150 km length by the duration of beach-ridge formation. Assuming an average thickness of 10 ±2 m, the overall average accumulation rate over the period 1800 BCE until today has been 2.3–3.5 million $m^3$/yr. Accumulation rates along Transects A, B and C range between 16 and 54 $m^3$/m/yr (Table 2).

The calculated average accumulation rate is exceptionally high compared to those reported for other large beach-ridge systems, such as 0.05 million $m^3$/yr at Guichen Bay, Australia (Bristow and Pucilllo, 2006), 0.14 million $m^3$/yr at Keppel Bay, Australia (Brooke et al., 2008a) and 1.7 million $m^3$/yr at Kujukuri, Japan (Tamura et al., 2010). As these systems are much shorter than the Usumacinta-Grijalva plain, accumulation rates are more similar when expressed in $m^3$/m/yr. For two other large beach-ridge systems with detailed chronological control we estimate accumulation rates of 0.92 million $m^3$/yr (Nayarit, Mexico; using cross sections in Curray et al., 1969), and 1.4 million $m^3$/yr (Katwijk, the Netherlands; using sections in Cleveringa, 2000).

Average aeolian accretion rates along Transects A, B and C range between 1.2 and 6.1 $m^3$/m/yr (Table 2), with relatively high values along Transect B during Phase 2 and along Transect A during Phase 3B. Rates are much higher than the average long-term aeolian accretion rates of 0.1 – 0.6 $m^3$/m/yr for three beach-ridge plains in southeastern Australia (Oliver, 2016) but are relative low compared to average long-term accretion rates for larger-scale foredunes, which roughly vary between 5 and 20 $m^3$/m/yr (e.g. Aagaard et al., 2004; Ollerhead et al., 2013; Keijsers et al., 2014).

Aeolian accretion rates are ca. 5-20% of the total volumetric growth rate of the beach-ridge plain (Table 2), comparable to the 10.5% inferred for the Moruya beach plain, Australia (Oliver, 2016). Aeolian processes therefore play a minor role in beach-plain sediment accretion. We found a relatively large contribution of aeolian accretion (20-30% of total beach-ridge accretion) for beach ridges formed along Transect B between approximately 1800 BCE and 30 CE (Phase 2), which could be an indication of stronger easterly trade winds during this time.



### 4.8 Evolution of the beach-ridge plain

The new chronological, geomorphological and sedimentary data enabled us to reconstruct the three-phased development of the beach ridge complex in considerably more detail than previous researchers.

The oldest part of the beach-ridge sequence (Phase 1) has been most completely preserved on the inland side of the barrier complex, southwest of the current confluence of the Grijalva and Usumacinta Rivers (Tres Brazos, Fig. 2b). Here, beach ridges are partly covered by organic-rich back-barrier marsh deposits that locally reach thicknesses of up to 4 m (e.g. core 307; Fig. 3b). To the east of Tres Brazos (Fig. 2b), no Phase 1 beach-ridge topography is discernible from the DEM. Any Phase 1 ridges were likely eroded over time by the migrating Usumacinta River. Our oldest age of 4248 ±90 BCE (at 1σ) for freshwater organic deposits (sampled in core 307, Fig. 3b), post-dates the onset of coastal progradation in the study area. This organic unit formed after the oldest beach ridges had developed, suggesting that the inception of the Usumacinta-Grijalva beach-ridge plain (i.e. the onset of Phase 1), marking the transition from transgressive to regressive conditions, probably occurred centuries earlier (ca. 4500 BCE).

Relatively coarse-grained beach ridges, inferred to be supplied with sediment by a branch of the Grijalva River, accreted during Phase 1A along the inland part of Transect B (Fig. 5a). This set of beach ridges formed until 2800 BCE, at a time when RSL was several meters lower than today. Nowadays, only the most elevated beach ridges formed during that phase protrude from the marshy plain.
During Phase 1B, which lasted until 1800 BCE, the Usumacinta River system increasingly supplied relatively fine sediment to the area, as its SP y SP distributary developed. The inland part of Transect A shows that the new promontory at the mouth of the SP y SP did not immediately developed the characteristics of a mature beach-ridge plain. At core location PP1 and at Pozpetr (Fig. 3a), only clayey estuarine and organic flood-basin deposits occur. The first beach-ridge sand body only starts near core 336. The few linear structures in the DEM that are discernible further inland may represent chenier-like features (as tentatively indicated in Fig. 3a). The Grijalva River system continued to influence beach-ridge formation in the area of Transect B. During Phase 1B it made use of the 'Popal Grande palaeochannel' (cf. Psuty, 1967), which was active between approximately 2800 and 2100 BCE (Fig. 5).

During Phase 2 (1800 BCE – 150 CE), the SP y SP promontory further developed. Its relative large acute angles between beach ridges and the present-day coastline (Fig. 2), indicate that riverine sediment supply contributed significantly to the growing beach ridge complex. Fluvial contributions from more easterly sources are improbable, because sizeable rivers have not been present east of the SP y SP branch. In addition, calcareous biogenic sediments dominate in that sector of the coastal-lagoonal plain, particularly east of Ciudad del Carmen (Fig. 1b). A marine source area is unlikely as well, because surface sediments in front of the SP y SP river mouth are predominantly composed of clay and fine silt (Ayala-Castañares and Guttiérrez-Estrada, 1990). A possible marine source area for beach-ridge sands is the seabed in the western part of the study area (Fig. 1b), but there is no known mechanism that could have moved vast amounts of sediment against the dominant drift direction. A terrestrial contribution via longshore current, sourced from the Grijalva River mouth, is unlikely for the same reason: the necessary transport path would be opposite the dominant drift direction. Moreover, the main distributaries of the Grijalva River system at the time were positioned farther westward than at present (e.g. the Pajonal and Blasillo palaeodistributaries described by Von Nagy (2003) (Fig. 1b). Towards the end of Phase 2, a slight increase in acute angles of the beach ridges is seen about 5 km west of the present main outlet (Fig. 2a). This local anomaly from the overall pattern indicates temporal



activation of a distributary river mouth at this location, which may be seen as a precursor of the
nearby main outlet active during Phase 3.
The break between Phases 2 and 3 is set at the marked increase in beach-ridge elevation, and at
regionally truncated beach ridges in the area near Transect B. These features indicate a major
reorganisation in the Grijalva and Usumacinta distributary network and river mouths. Around
150 CE, a major new delta promontory began to develop, that still is the joint outlet of the
Grijalva and Usumacinta rivers today. In its development, we distinguish three sub-phases.
During Phase 3A, the old SP y SP outlet was still functioning. At the end of Phase 3A, the
Usumacinta had fully avulsed towards its current location, terminating sediment delivery at the
old outlet. The age-distance model of Transect A2 (Fig. 4c) indicates that this latter avulsion
occurred around 1050 CE. The break between Phases 3A and B is marked by a shift in beach-
ridge orientations. West of the SP y SP abandoned outlet, elevated beach ridges are related to
increased sediment supply due to cannibalisation of the former promontory. Even today, the old
SP y SP promontory is still eroding, with current rates around 3.5 m/yr (Ortiz-Pérez, 1992; Ortiz-
Pérez et al., 2010).
The break between Phases 3B and 3C, placed at 1460 CE, is not related to river-outlet
repositioning and therefore morphometrically more arbitrary. It is reflected by moderate
increases in progradation rate (Table 2).

## 5 Discussion

**5.1 Beach-ridge-formation model**

Psuty (1965, 1967) suggested an important contribution of storm surges and related overwash to
the development of the Usumacinta-Grijalva beach ridges. Our GPR measurements revealed only
evidence for swash-built beach ridges with an aeolian cap on top, whereas typical landward-
dipping reflections from washovers have not been identified. In addition, the sandy deposits do
not include any exceptionally coarse sand layers within the upper part of the cores, and most of
the analysed sand samples from above MSL were characterised as aeolian in origin. The DEM of
the area shows little evidence of extreme storm events impacting the area; scour holes were only
identified along one beach ridge, formed around 1450 CE. Nevertheless, storms do play a role in
beach-ridge formation. Strong north-westerly winds during 'Nortes', for example, cause beach
erosion (West et al., 1969). Owing to a temporal reversal in the longshore-current direction, sand
is transported eastward and contributes to beach-ridge formation several months after the storm
event. Individual storms associated with the nearby passage of hurricanes will also lead to beach
erosion. In both cases foreshore recovery likely takes places within a few months after the
erosional event (Carter, 1986 and references therein).
The GPR data show that each beach ridge in the study area likely starts as a wave-built swash
bar, formed over a period of 7 - 19 years. Once stabilised and no longer subject to hydrodynamic
processes, subsequent wind processes create an aeolian cap on the ridge. Sand is blown in from
the adjacent beach, including the active intertidal swash bar (exposed during low tide). It is
trapped by pioneer vegetation, especially *Ipomoea pes-caprae*, that rapidly colonises the young
ridge. The final ridge elevation is determined by the length of the period that the ridge is located
next to the beach: the longer the ridge is exposed to aeolian sand deposition, the higher it
becomes. Consequently, high beach ridges arise when coastal propagation rate is low. Along
individual beach ridges, sections formed relatively close to an active river apex, where
progradation rates are high (Fig. 11b), are lower than those formed farther away (Fig. 11c),
where progradation rates are low. Apparently, reduced sediment supply leads to higher ridges.



## 5.2 Beach-ridge elevation as a proxy of riverine sediment supply

Beach-ridge elevation is negatively correlated with progradation rate, both in shore-normal
(Transect A, Phase 3A, Fig. 4c) and in a longshore direction (Fig. 11c). For periods when rivers
supplied most of the sediment stored in the beach-ridge system, we hypothesise that ridge
elevation along shore-normal transects may be used as a proxy of fluvial sediment supply
through time and space. Owing to the large storage capacity within the river basin, sediment
availability for fluvial transport is not a limiting factor. Peak river-discharge events and extended
periods of large supply translate into high progradation rates and lower ridges. Periods of
reduced supply during dry conditions, when rivers are less capable of transporting large amounts
of sand, result in higher ridges. Evidence for our hypothesis is provided by a comparison of the
beach-ridge morphology with independent information on climate in the catchments. We found
relatively high beach ridges along sections of Transects A, B and C formed during the period
between 810 – 950 CE (Fig. 10). This period, associated with the Maya Classic collapse, is well
known for the occurrence of multiple prolonged droughts in southern Mexico (cf. Hodell et al.,
1995) and Guatemala (cf. Wahl et al., 2014).

Direct sediment supply by rivers, however, is not always the main driver in coastal progradation.
Cannibalisation of abandoned promontories may generate abundant sandy sediment for
anomalously high sediment supply along the downdrift beach. A drastic increase in sediment
supply due to the erosion of the SP y SP promontory after the avulsion of the Usumacinta River
around 1050 CE resulted in increased availability of sand for aeolian reworking, triggering the
formation of relatively high beach ridges on both sites of the eroding SP y SP promontory (Figs.
4b and 10). Even ~1000 years after the avulsion that caused the Usumacinta River to join the
Grijalva River at Tres Brazos, coastal erosion at its former SP y SP apex is still ongoing. This
process is obscuring the relationship between direct fluvial sediment supply and beach-ridge
elevation, but can be recognised as a separate force because it caused major changes in geometry
and orientation of beach ridges (Fig. 2a), as well as clear changes in grain-size characteristics
(Figs. 4b and 5b).

Detecting changes in fluvial sediment supply from beach ridge elevation differences requires that
there are no major changes in wave and wind climate affection the signal. Such changes in wave
and wind climate should be reflected in significant changes in the granulometric parameters of
the deposited beach ridge sand. After normalising for the effects of new river-mouth initiation
and old promontory abandonment, we find only minor remaining granulometric differences in
our study area. Comparison of modern deposits to the fossil beach deposits of Transect A (Fig.
B2) suggests that wind and wave climate (multi-decadal averaged) during the past 2000 years
(Phase 3) have been comparable to those of the present. In contrast, the different geometry of the
beach-ridge plain formed during the earlier Phase 2 (Fig. 11a) indicates that wind and wave
climate at that time were likely different from the situation today. During Phase 2, progradation
rates decreased relatively slowly with increasing distances from the SP y SP River mouth (Fig.
11b), and the promontory was less asymmetric than the promontory formed during Phase 3C at
the joint outlet of the Usumacinta and Grijalva Rivers. This difference can be explained by a
higher contribution of high-angle waves from the west in the construction of the delta
promontory, especially over the past 500 years, which is in agreement with model simulations of
delta development near river outlets (Ashton and Giosan, 2011). Such geometric changes can
thus occur without changes in sediment supply.
We speculate that the increased contribution of high-angle waves during Phase 3 is a possible
response to the increasingly frequent occurrences of north-westerly winds, probably related to a
stronger and more frequent contribution of cold fronts than before. During Phase 2, the

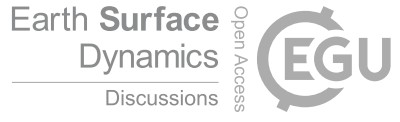

Intertropical Convergence Zone (ITCZ) was farther northward, and likely associated with
stronger easterly trade winds that could have caused the westward increase in aeolian accretion
rates during this time period (Fig. 11c).
**5.3 Beach-ridge periodicity**
Combining the age-distance modelling with the LiDAR-derived beach-ridge morphometrics
(Fig. 2a), it is evident that the development of past ridge-swale couplets took between 7 and 19
years (Table 2), and that the time interval for the formation of subsequent ridge-swale couplets
decreased with increasing progradation rate (Table 2). This relationship is apparent not only in
shore-normal transects marked by variable progradation rates, but also in a shore-parallel
direction, with beach ridges merging away from the river mouth supplying the sediment. It
corroborates a similar finding of Thompson (1992) for Lake Michigan beach ridges and indicates
that an allogenic cause of individual beach-ridge formation (e.g. periodic decimetre-scale lunar
or steric sea-level oscillations; Tanner, 1995), is unlikely. In this light, it should be noted that
long time series of water-level data from seven tide gauges along the southern Gulf of Mexico
(Salas-de-León et al., 2006) do not show any decadal periodicity. The inter-annual amplitude
variability is only a few centimetres, an order of magnitude lower than the intra-annual
amplitude range of 25 cm between a February low and an October high. We therefore conclude
that ridge-swale couplets at the study site are not formed in response to RSL oscillations. This
finding agrees with the findings of Tamura (2012) and Moore et al. (2016) that the formation of
individual ridge/swale couplets is driven by autocyclic processes (Moore et al., 2016).
Comparison with periodicities reported from other large beach-ridge systems (Fig. 12) indicates
that low periodicities (< 25 yr) indeed are generally found at sites with high progradation rates
(>1.5 m/yr).
**6 Conclusions**
Our study demonstrates the importance of riverine sediment supply in the formation of the
Usumacinta-Grijalva beach-ridge sequence, corroborating earlier geomorphological studies
(Psuty, 1965, 1967; West et al., 1969). In contrast to this earlier work, we propose a mechanism
of ridge formation without a significant role of storm surges and over-wash deposits. The fine
sandy beach ridges were mainly swash built, have an aeolian cap, and likely formed under fair-
weather conditions without the requirement of sea-level oscillation. Autocyclic processes
controlled the periodicity (7-19 yrs) in beach ridge formation. The relatively low periodicities are
related to high progradation rates (> ~1.5 m/yr) and reflect ample sediment supply. The
indicative meaning of beach-ridge periodicities in palaeoenvironmental reconstructions is
limited.
We estimate that sediment supply, distributed along 150 km of coastline, was roughly 2.3 – 3.5
million m$^3$/yr, which is exceptionally large compared to that of other large beach-ridge
sequences. This can be attributed to extensive availability of easily erodible Los Chocoyos
ignimbrites in the headwater catchments of the Usumacinta River, given the abundance of
fragmented volcanic material derived from this unit in the beach ridge sands.
Our observations enabled us to subdivide the three main phases in the development of the beach-
ridge plain (Psuty, 1965, 1967; West et al., 1969) further into six sub-units, related to changes in
the configuration of the main river distributaries of the Usumacinta and Grijalva River system.
Combined $^{14}$C and OLS dating provided a robust and consistent chronological framework for




these phases, which substantially improved the existing chronology based on radiocarbon-dated
shell material (Aguayo et al., 1999).
Our analyses show that during periods when the Usumacinta River was the main supplier of
sandy sediments to the coast, changes in river discharge determined sediment availability,
progradation rate, and the final elevation of the beach ridges. Since the river discharge is directly
related to rainfall in the river catchment, beach ridge elevation may be an excellent proxy for
temporal changes in regional-scale precipitation.

## Acknowledgements

We thank INEGI, Mexico for the generous provision of the LiDAR data, and Hector V. Cabadas-
Báez for kindly supplying the major-element data of glass shards recovered from levee deposits
at Tierra Blanca. Elise van Winden, Jesse Hennekam and Ryan Nagelkirk provided field support
and Salomon Kroonenberg offered valuable advice. This research is supported by the
Netherlands Organisation for Scientific Research (NWO grant 821.01.007). Remke van Dam
acknowledges support from the Michigan Space Grant Consortium.

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



**Figure captions**
Figure 1: (a) Location of the Usumacinta-Grijalva beach-ridge sequence (yellow) along the edge
of the Holocene delta plain (blue) and the drainage basins of the two main rivers traversing the
headlands of this delta (red outlines). Simplified geological map modified from Garrity and
Soller (2009) and extent of Los Chocoyos pyroclastic flow deposits adopted from the geological
map of Guatemala at scale 1:500,000 (Instituto Geográfico Nacional, 1970; Koch and McLean,
1975; Rose et al.,1987; and Sánchez-Núñez et al., 2015). Elevated uplands above 500 m+MSL,
outlined using the SRTM 1-arc-second dataset (USGS, 2009), are depicted in gray; (b) Overview
of the Usumacinta-Grijalva delta and the three main phases of Holocene beach-ridge formation
defined by Psuty (1965, 1967). The apexes of the two main rivers (yellow dots) are indicated
with 25, 50 and 75 km equidistant lines (red lines). Nearshore distribution of coarse silty to
gravelly surficial sediments after Ayala-Castañares and Guttiérrez-Estrada (1990). Surficial
sediments from the remaining part of the continental shelf are composed of clay and fine silt.
Figure 2: (a) LiDAR-based DEM and location of studied transects, with the GPR transects in
blue; (b) Main beach-ridge-formation phases, and locations of sediment cores (black) and of
samples collected for OSL and AMS $^{14}$C dating. Numbers 1-15 denote the fifteen cross-normal
ribbon-shaped elevation transects, in the text referred to as B2-1, B2-2, etc.
Figure 3: Age-distance models for Transects A (a), and B (b). Indicated are the 1 sigma
distribution for the model results using the P_sequence module in Oxcal 4.2 (Bronk Ramsey,
2009). Sample locations of AMS $^{14}$C (black squares) and OSL (red dots) samples are indicated,
and projected samples are presented in italics. The calibrated $^{14}$C ages are indicated with the full
probability distribution and the OSL ages (red and yellow triangles) with their 1 sigma range.
CaCO$_3$ content for selected core samples indicates pedogenic decalcification depth, used to
estimate the position of MSL during beach-ridge formation. The dashed trendline is based on
Gischler and Hudson's (2004) reconstruction of late-Holocene RSL.
Figure 4: (a) Core locations along Transect A2; (b) Median grain size of analysed sand samples,
with associated shoreface-dipping angle; (c) Age-distance model (after Nooren et al., 2017) and
OSL ages (red dots)  (with 1 sigma probability).
Figure 5: (a) Reconstructed palaeoshorelines (ages in Year CE); (b) Median grain size (μm) of
wave-formed and aeolian deposits (large and small dots, respectively).
Figure 6: (a) Shore-parallel variability in grain-size parameters of swash (red) and aeolian
(black) facies. Vfs = very fine sand; fs = fine sand; ms = medium sand; cs = coarse sand.
Figure 7: (a) Grain-size variability along a beach-to-nearshore profile of surficial grab samples
taken during fair-weather conditions in April 2013 at Playa Estrella (see Fig. 6 for location).
Sand characteristics of beach core 197 (triangle; sample from –3.5 m+MSL), taken in 2012 at the
same location are shown for comparison. The sand sample likely had its origin in the nearshore
at a distance of 240 - 300 m from the contemporary low-tide line (here shown at 240 m),
assuming a comparable beach profile during time of deposition. LWL and HWL are mean low
and high water level; (b) Grain-size distribution of representative surficial sand samples from the
beach profile, denoted by coloured circles in Fig. 7a.



Figure 8. Processed GPR data and interpretation for two closely spaced relatively elevated beach ridges along Transect A (see Fig. 2a and 3a for location), collected using a GSSI system with 250MHz shielded antennas. Processing steps included signal dewow to remove low-frequency content, a custom gain function to amplify deeper reflections, background removal below the direct waves, and topographic correction. Time-to-depth conversion for the elevation axis was based on velocities of 0.125 and 0.06 m/ns above and below the water table, respectively. The position of the water table at 0.2 m+MSL  (blue dashed line) was drawn on the basis of changes in reflection characteristics, and confirmed by observations from core 72 (black arrow). Here, the water table was positioned at 2.2 m below the land surface. Highlighted in the interpretation are foreshore and shoreface deposits (black dipping lines), the transition from foreshore to backshore and/or aeolian deposits at 0.8 m+MSL (red dashed line), and some landward-dipping structures (orange lines) possibly related to infill of a large former runnel. The curved reflections around 110 m (40 ns and deeper) are caused by surface scattering off a large nearby tree that was passed while moving the GPR along the transect.

Figure 9: $SiO_2$ - CaO diagram for analysed volcanic glass shards, plotted along with compositional characteristics of El Chichón (Nooren et al., 2017) and Los Chocoyos tephra (Kutterolf et al., 2008). Data points represent averages for 5-12 particles (bars are 1 sigma). The $SiO_2$ – CaO composition of volcanic glass shards recovered from Usumacinta levee deposits at Tierra Blanca III (Cabadas-Báez et al., 2017) are indicated for comparison. We refer to table A3 for all major element data. Inset: Thin section of pumice and volcanic glass shards recovered from the beach-ridge sands (core 197, sample from 80 cm below surface). Notice elongated vesicularity of one of the pumice fragments.

Figure 10: Mean beach-ridge elevation variability along shore-normal Transects B (a), A (b) and C (c). See Fig.11a for the location of the individual transects. Notice relatively high beach-ridge elevations around 800-950 CE for all three transect. This period is known for the occurrence of multiple prolonged droughts, and has been related to the Classic Maya collapse.

Figure 11: Variability in shore-parallel beach-plain progradation rate (b) and mean elevation (c) for Phase 2 (1800 BCE - 150 CE) (orange/red) and Phase 3C (1460 - 1965 CE) (green). Dashed lines represent calculated elevation values for constant 'aeolian' accretion rates. Arrows in panel (a) indicate the estimated dominant direction of swell driving the formation of the swash deposits, and the dominant wind direction related to aeolian sand transport, responsible for the formation of an aeolian cap on top of the swash-built beach ridges.

Figure 12: Periodicities of beach-ridge formation for the Usumacinta-Grijalva (Us-Gr) system compared with reported or estimated values for other large beach-ridge systems: Rockingham Bay (Forsyth et al., 2010), Beachmere (Brooke et al., 2008b), Moruya (Oliver et al., 2015), Guichen Bay (Murray Wallace et al., 2002; Bristow and Pucillo, 2006), Keppel Bay (Brooke et al., 2008a), Shark Bay (Nott, 2011), Cowley beach (Nott et al., 2009), Lake Michigan (Thompson, 1992), Sint Vincent Island (Lopez and Rink, 2008; Rink and Lopez, 2010), Jerup (Nielsen et al., 2006), Nayarit (Curray et al., 1969) and Rio Grande do Sul (Milana et al., in press).

Table 1. General characteristics for the watersheds of the main rivers draining towards the Usumacinta-Grijalva delta.

Table 2: General characteristics of the beach-ridge plain along the shore-normal transects as indicated in figure 2A.






## Appendix A



Table A1: AMS [14]C-dated samples.

Table A2: OSL-dated samples

Table A3: Major-element composition (mean and standard deviation) of volcanic glass and
pumice fragments recovered from the beach-ridge sediments along Transect A. Oxide
concentrations are normalized to 100% on a volatile-free basis. All iron is taken as FeO. The
major-element composition of volcanic glass shards from Tierra Blanca III were generously
provided by Hector V. Cabadas-Báez (Cabadas-Báez et al., 2017).

## Appendix B



Figure B1: Variability in grain-size distribution of sand samples along Transect A at 0.04-14.5
km from the current coastline. Vfs = very fine sand; fs = fine sand; ms = medium sand; cs =
coarse sand. Grain-size distributions of representative surficial samples from the current beach
profile (Fig. 7c) are indicated for comparison.

Figure B2: Age-distance scenarios for Transect B2, assuming a constant aeolian accretion rate in
a shore-normal direction. The combined calibrated ages for OSL and AMS samples 440 and
433/336 (154 +/-65 and 1720 +/-65 BCE), calculated with Oxcal 4.2 (Bronk Ramsey, 2009)
using the IntCal13 calibration curve (Reimer et al., 2013), are used as model boundaries.
Indicated are five long-range (red) and five short-range (blue) scenarios for Transect B2-1 – B2-
5. The calibrated 1 sigma age range for a P_sequence model solely based on OSL ages
(excluding sample 437) is indicated in grey.





Figure 1



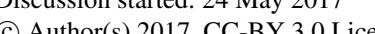


Figure 2





Figure 3a



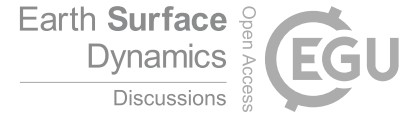

Figure 3b



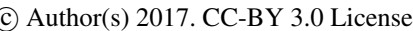


Figure 4.





Figure 5





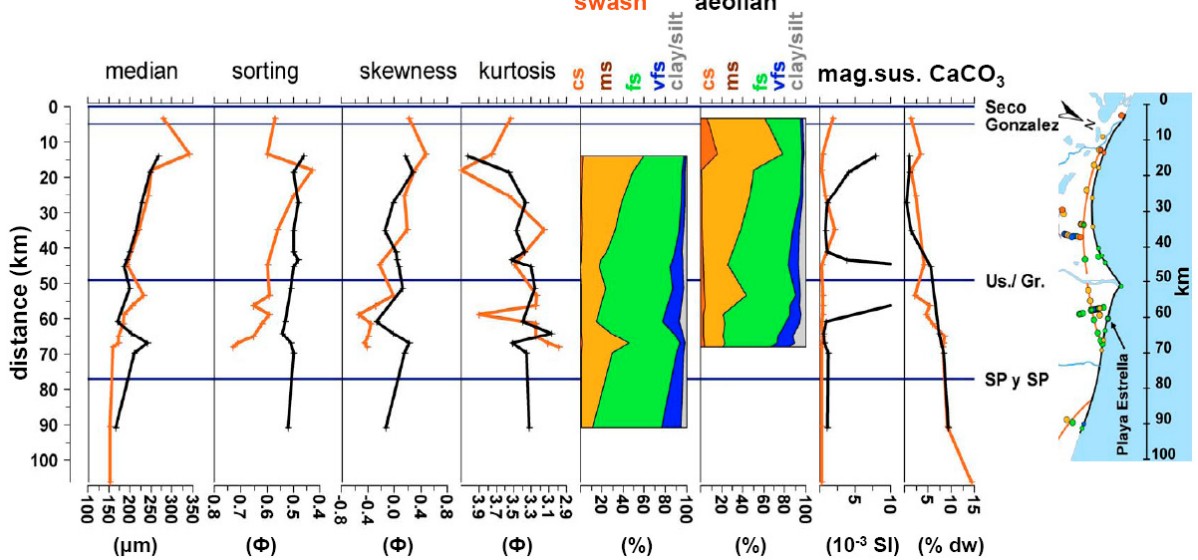

Figure 6.

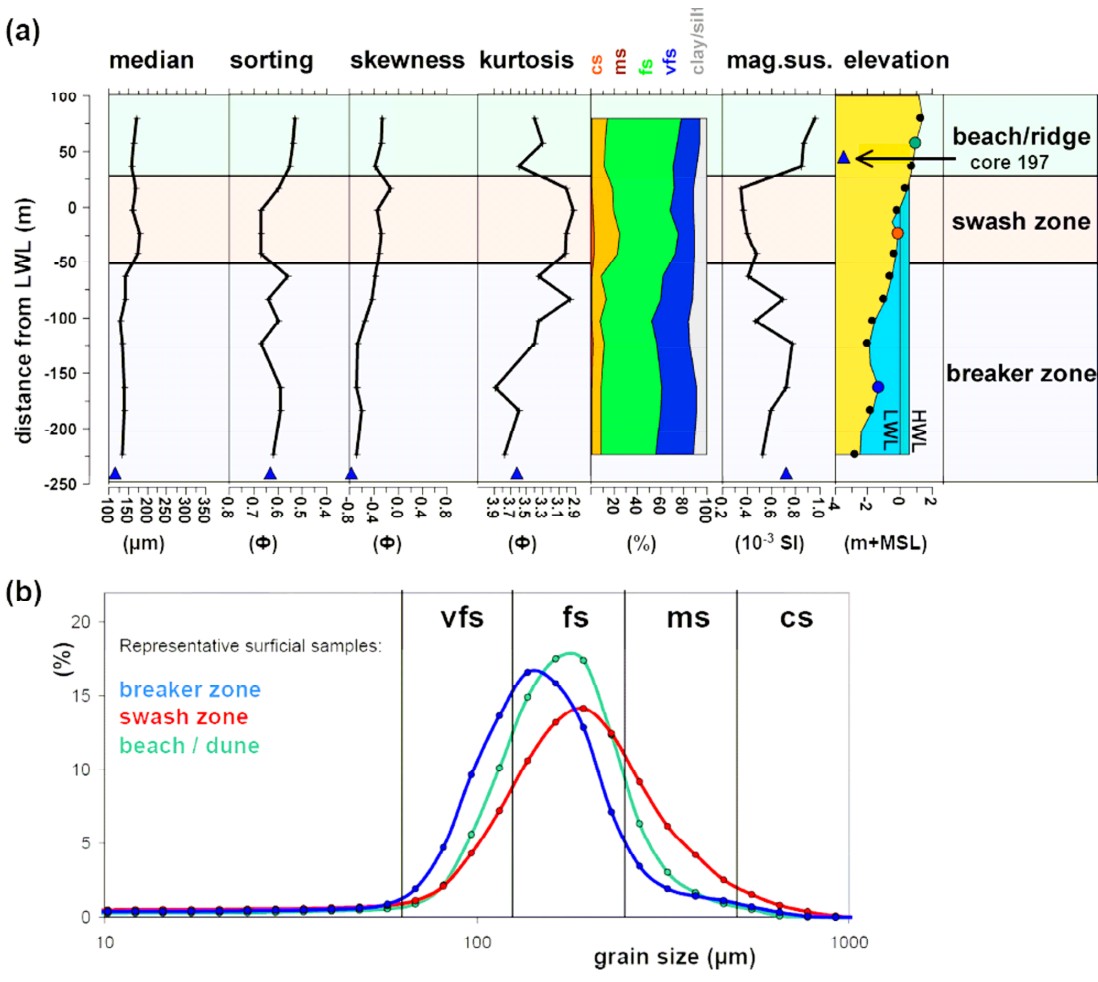

Figure 7



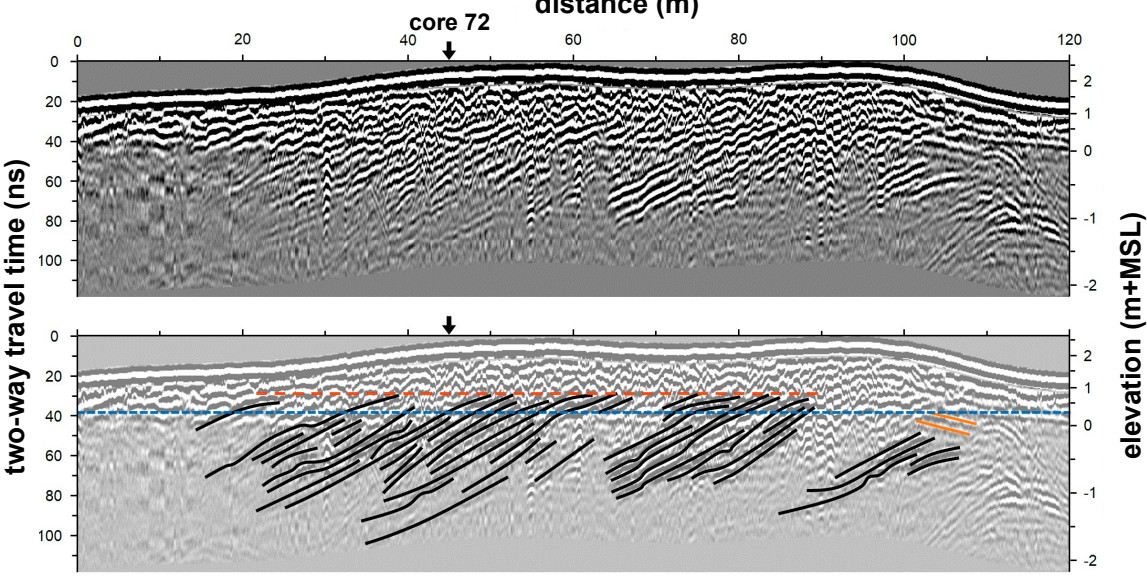

Figure 8

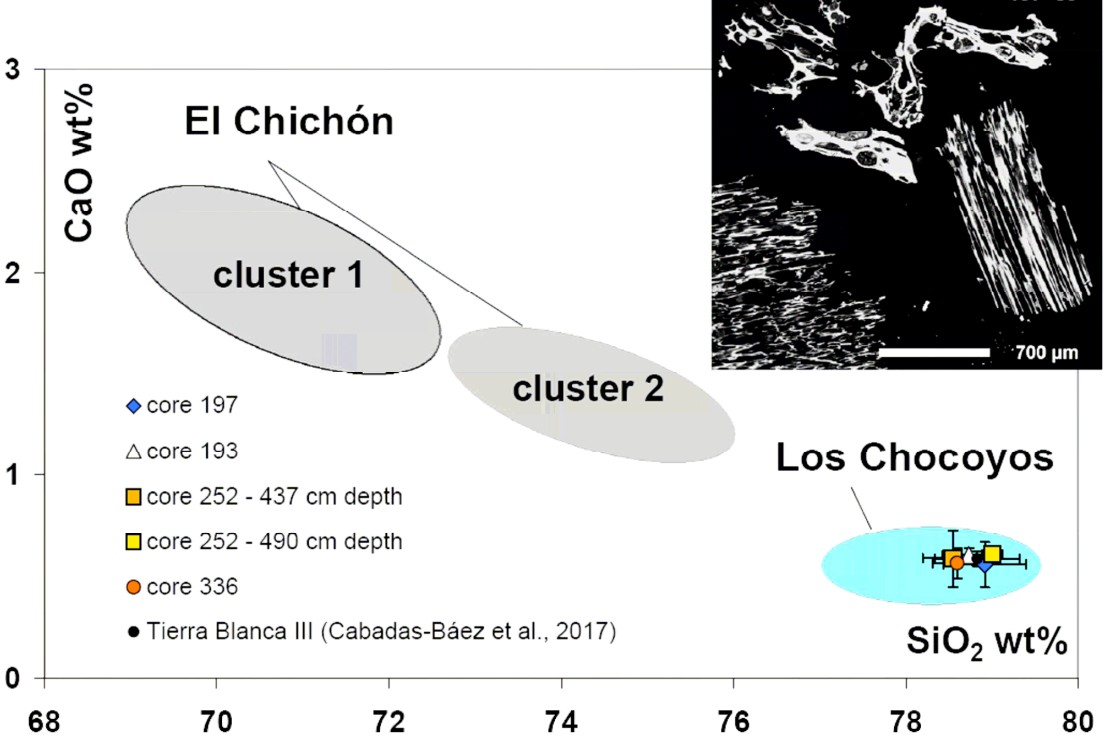

Figure 9.





Figure 10.






Figure 11.



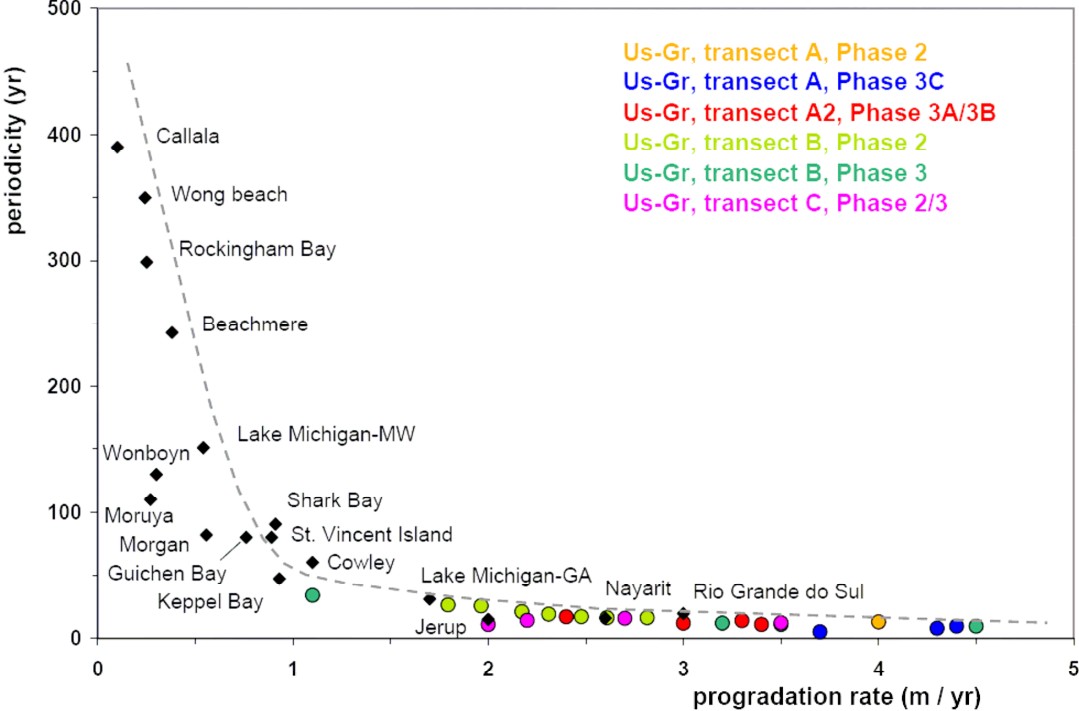

Figure 12

Table 1.

| catchment | areal extent | | average annual precipitation | average annual discharge | excess rainfall |
|---|---|---|---|---|---|
| | (km2) | (%) | (mm/yr) [1] | (m³/s) [2] | (%) |
| Usumacinta | 70714 | 58 | 2150 | 2000 | 41 |
| Grijalva | 37471 | 31 | 1500 | 750 | 42 |
| Sierra/Chilapa system | 12840 | 11 | 2550 | 600 | 58 |

1. mean value for the watershed, calculated over the period 1950-2000
   (WorldClim version 1.4 (release 3); Hijmans et al., 2005)
2. estimated valued based on measured discharges at the different hydrological stations
   (Banco Nacional de Datos dee Aguas Superficiales, consulted in June 2016)

Table 2.

| phase | time range Year CE | transect | distance (km) | duration (yr) | mean elevation (m+MSL) | number of ridges | mean periodicity (yr) | mean progradation rate (m/yr) | total beach plain accretion (m³/m/yr)[1] | mean aeolian accretion (m³/m/yr) | (%) |
|---|---|---|---|---|---|---|---|---|---|---|---|
| 3c | 1460 - 2007 | A | 2.5 | 547 | 1.5[2] | 58 | 9.4 | 4.5 | 36 - 54 | 4.3 | 8 - 12 |
| | 1460 - 2007 | B3 | 2.5 | 547 | 1.41 | 57 | 9.6 | 4.5 | 36 - 54 | 4 | 7 - 11 |
| | 1460 - 2007 | C | 1.1 | 547 | 1.37 | 50 | 10.9 | 2.0 | 16 - 24 | 2.6 | 11 - 16 |
| 3b | 1050 - 1460 | A | 1.6 | 410 | 2.1[2] | 59 | 6.9 | 4.0 | 32 - 48 | 6.1 | 13 - 19 |
| | 1050 - 1460 | B3 | 1.4 | 410 | 1.49 | 37 | 11.1 | 3.5 | 28 - 42 | 3.6 | 9 - 13 |
| | 1050 - 1460 | C | 0.9 | 410 | 1.31 | 29 | 14.1 | 2.2 | 17 - 26 | 2.7 | 10 - 16 |
| 3a | 150 - 1050 | A | 2.4 | 900 | 1.55[2] | 61 | 14.8 | 2.7 | 21 - 32 | 2.7 | 8 - 13 |
| | 150 - 1050 | B3 | 1.9 | 900 | 2.03 | 47 | 19.1 | 2.1 | 17 - 25 | 2.9 | 12 - 17 |
| | 150 - 1050 | C | 3.1 | 900 | 0.78 | 74 | 12.2 | 3.5 | 28 - 42 | 2.6 | 6 - 9 |
| 2 | -1800 - 150 | A | 7.8 | 1950 | 1.44[2] | 150 | 13.0 | 4.0 | 32 - 48 | 3.6 | 8 - 11 |
| | -1800 - 150 | B2 | 4.7 | 1950 | 2.21 | 120 | 16.3 | 2.4 | 19 - 29 | 5.8 | 20 - 30 |
| | -1400 - 150 | C | 4.3 | 1550 | 0.47 | 98 | 15.8 | 2.7 | 22 - 33 | 1.2 | 4 - 5 |
| 1 | 4300 - 1900 | B1 | 11.1 | 2400 | 0.82 | 154 | 15.6 | 4.6 | | | |

1. assuming an average thickness for the beach ridge deposits of 8 - 12 m
2. 1 m was added to the LiDAR-elevation data from 2008





# Appendix A

Table A1

| | sample | dist. along transect (m) | GrA | Age BP | sigma | extracted fraction | d¹³C (°/₀₀) | C (%) |
|---|---|---|---|---|---|---|---|---|
| **TRANSECT A** | | | | | | | | |
| | **debris layers within beach ridge sands** | | | | | | | |
| | 429-250L | 2110 | 58037 | 300 | 35 | leaf fragments | -27.39 | 49.78 |
| A2[1] | 393-300L | 3120 | 58032 | 715 | 35 | leaf fragments | -30.25 | 45.26 |
| A2 | 390-330L | 3375 | 59436 | 755 | 30 | leaf fragments | -29.45 | 49.48 |
| A2 | 389-330L | 3485 | 58031 | 900 | 40 | leaf fragments | -30.48 | 52.21 |
| A2 | 386-240L | 3665 | 59755 | 820 | 40 | leaf fragments | -31.97 | 51.93 |
| A2 | 386-610L | 3665 | 59751 | 940 | 50 | leaf fragments | -30.52 | 49.28 |
| A2 | 381-225L | 4195 | 59753 | 935 | 35 | leaf fragments | -31.12 | 55.98 |
| A2 | 379-280L | 4375 | 58030 | 1015 | 35 | leaf fragments | -28.29 | 48.16 |
| A2 | 378-280L | 4475 | 59435 | 990 | 30 | leaf fragments | -28.95 | 52.84 |
| A2 | 376-290L | 4710 | 59752 | 1075 | 40 | leaf fragments | -28.29 | 42.98 |
| A2 | 193-171L | 4890 | 55022 | 1250 | 30 | leaf fragments | -30.05 | 61.50 |
| A2 | 196-204L | 4978 | 55023 | 1235 | 30 | leaf fragments | -30.94 | 61.50 |
| A2 | 396-270L | 5330 | 59757 | 1255 | 40 | leaf fragments | -30.94 | 51.96 |
| A2 | 397-350L | 5415 | 58033 | 1390 | 35 | leaf fragments | -30.06 | 54.93 |
| A2 | 398-260S | 5520 | 59437 | 1270 | 30 | squash seed | -29.45 | 49.48 |
| A2 | 413-270L | 5595 | 59438 | 1415 | 30 | leaf fragments | -28.72 | 50.93 |
| A2 | 400-295L | 5700 | 59694 | 1775 | 40 | leaf fragments | -30.31 | 45.30 |
| A2 | 481-290L | 5755 | 60873 | 1490 | 35 | leaf fragments | -29.65 | 51.11 |
| A2 | 480-290L | 5790 | 60871 | 1525 | 35 | leaf fragments | -29.74 | 51.38 |
| A2 | 426-885L | 5935 | 58035 | 1665 | 35 | leaf fragments | -29.92 | 45.92 |
| A2 | 426-255L | 5935 | 58034 | 1690 | 40 | leaf fragments | -29.75 | 52.56 |
| | 252-485L | 8642 | 55021 | 2420 | 35 | leaf fragments | -31.42 | 55.10 |
| | 252-485C | 8642 | 55024 | 3290 | 30 | charcoal | -24.66 | 73.70 |
| | 336-368L | 14222 | 54940 | 3410 | 45 | leaf fragments | -29.7 | 38.70 |
| | 336-368C | 14222 | 55025 | 3990 | 35 | charcoal | -25.02 | 68.20 |
| | **base of freshwater peat** | | | | | | | |
| | Pozpetr.-78-82[2] | | UtC-11090 | 2055 | 59 | charcoal/wood | -28.2 | |
| | PP1-169-170[1] | | 53751 | 3220 | 40 | charred plant fragments | -21.55 | 53.40 |
| | **base of mangrove peat** | | | | | | | |
| | LC1-315-320 | | 55026 | 5030 | 35 | charred plant fragments | -23.78 | 79.20 |
| **TRANSECT B** | | | | | | | | |
| | **debris layers within beach ridge sands** | | | | | | | |
| | 443-230L | 1075 | 58041 | 165 | 35 | leaf fragments | -28.11 | 50.18 |
| | 444-150L | 2270 | 58042 | 350 | 35 | leaf fragments | -28.64 | 49.99 |
| | 446-275L | 4134 | 58043 | 1060 | 40 | leaf fragments | -29.53 | 52.03 |
| | 440-350L | 6168 | 58040 | 2125 | 40 | leaf fragments | -29.82 | 50.59 |
| | 185-471L | 7195 | 55029 | 2665 | 35 | leaf fragments | -28.61 | 42.30 |
| | 438-170L | 7752 | 58039 | 3005 | 35 | leaf fragments | -29.64 | 52.59 |
| | 188-310L | 10468 | 55020 | 3930 | 35 | leaf fragments | -30.33 | 51.60 |
| | 432-300L | 10866 | 58144 | 3880 | 40 | leaf fragments | -30.65 | 51.86 |
| | **base of freshwater peat** | | | | | | | |
| | 307-405-410S | 21901 | 64320 | 5420 | 70 | Asteraceae seeds | -28.08 | |
| **TRANSECT C** | | | | | | | | |
| | 469-160L | | 58044 | 1210 | 35 | leaf fragments | -29.63 | 49.70 |
| | 469-325L | | 58048 | 1360 | 35 | leaf fragments | -29.51 | 46.92 |

1) Nooren et al., 2017
2) Nooren et al., 2009

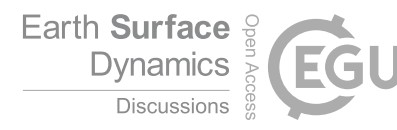

Table A2

| sample | NCL code | Lat. (°) | Long. (°) | comp. dist.[1] (m) | depth (m) | elev. (m+MSL) | water content meas. (% dw) | used (% dw) | ± | organic content (% dw) | ± | U²³⁸ (Bq/kg) | ± | Th²³² (Bq/kg) | ± | K⁴⁰ (Bq/kg) | ± | β (Gy/ka) | ± | γ (Gy/ka) | ± | cosmic radiation (Gy/ka) | ± | burial dose[2] (Gy) | ± | over-dispersion (%) | ± | dose rate (Gy/ka) | ± | age (ka) | ± | age (Year CE) | ± | validity |
|---|---|---|---|---|---|---|---|---|---|---|---|---|---|---|---|---|---|---|---|---|---|---|---|---|---|---|---|---|---|---|---|---|---|---|
| **Transect A** | | | | | | | | | | | | | | | | | | | | | | | | | | | | | | | | | | |
| 112 | NCL-4112227 | 18.595 | -92.594 | 3085 | 1.9 | 0 | 18.9 | 25 | 5 | 0.73 | 0.07 | 30.43 | 0.41 | 38.16 | 0.96 | 429 | 10 | 1.13 | 0.07 | 0.80 | 0.04 | 0.16 | 0.01 | 1.75 | 0.12 | 26 | 11 | 2.10 | 0.09 | 0.83 | 0.07 | 1182 | 70 | Likely OK |
| 427 | NCL-1114072 | 18.570 | -92.596 | 5280 | 2.3 | 0 | 27.7 | 25 | 5 | 1.26 | 0.13 | 13.67 | 0.17 | 14.71 | 0.37 | 724 | 15 | 1.39 | 0.09 | 0.65 | 0.03 | 0.15 | 0.01 | 3.6 | 0.2 | 30 | 4 | 2.20 | 0.1 | 1.65 | 0.12 | 363 | 120 | Questionable |
| 426 | NCL-4213072 | 18.568 | -92.595 | 5550 | 1.3 | 0 | 33.1 | 25 | 5 | 0.92 | 0.09 | 15.82 | 0.21 | 17.92 | 0.47 | 750 | 15 | 1.48 | 0.10 | 0.72 | 0.04 | 0.18 | 0.01 | 3.2 | 0.3 | 21 | 15 | 2.39 | 0.1 | 1.34 | 0.13 | 673 | 130 | Likely OK |
| 252 | NCL-4112229 | 18.549 | -92.575 | 7113 | 1.9 | 0 | 32.8 | 25 | 5 | 0.72 | 0.07 | 19.51 | 0.46 | 20.34 | 1.22 | 632 | 16 | 1.33 | 0.09 | 0.72 | 0.04 | 0.16 | 0.01 | 5.3 | 0.3 | 26 | 5 | 2.22 | 0.1 | 2.39 | 0.17 | -378 | 170 | Likely OK |
| **Transect B** | | | | | | | | | | | | | | | | | | | | | | | | | | | | | | | | | | |
| 443 | NCL-4213078 | 18.530 | -92.733 | 1075 | 1.6 | 0 | 25.9 | 25 | 5 | 0.87 | 0.09 | 20.50 | 0.32 | 23.05 | 0.76 | 577 | 13 | 1.25 | 0.08 | 0.71 | 0.04 | 0.16 | 0.01 | 0.5 | 0.04 | 20 | 8 | 2.13 | 0.09 | 0.24 | 0.02 | 1773 | 20 | Likely OK |
| 444 | NCL-1114071 | 18.522 | -92.726 | 2270 | 0.8 | 0 | 27.0 | 25 | 5 | 0.85 | 0.09 | 19.84 | 0.31 | 19.73 | 0.67 | 635 | 14 | 1.33 | 0.09 | 0.71 | 0.04 | 0.19 | 0.01 | 0.86[3] | 0.12 | 69 | 18 | 2.24 | 0.1 | 0.39 | 0.06 | 1623 | 60 | Likely OK |
| 445 | NCL-4213079 | 18.515 | -92.719 | 3255 | 1.05 | -0.05 | 25.0 | 25 | 5 | 0.84 | 0.08 | 14.83 | 0.27 | 15.26 | 0.68 | 725 | 15 | 1.41 | 0.09 | 0.67 | 0.04 | 0.18 | 0.01 | 1.79 | 0.08 | 7 | 6 | 2.28 | 0.1 | 0.79 | 0.05 | 1223 | 50 | OK |
| 179 | NCL-4112228 | 18.452 | -92.793 | 5750 | 0.55 | 1.55 | 4.7 | 5 | 5 | 1.78 | 0.18 | 18.87 | 0.30 | 23.12 | 1.58 | 612 | 13 | 1.60 | 0.09 | 0.86 | 0.05 | 0.20 | 0.01 | 2.9 | 0.3 | 33 | 9 | 2.66 | 0.1 | 1.08 | 0.12 | 932 | 120 | Questionable |
| 440 | NCL-4213077 | 18.463 | -92.761 | 6168 | 1 | 0 | 23.2 | 25 | 5 | 0.77 | 0.08 | 17.61 | 0.22 | 19.95 | 0.49 | 624 | 13 | 1.29 | 0.08 | 0.69 | 0.04 | 0.19 | 0.01 | 4.8 | 0.2 | 17 | 3 | 2.17 | 0.09 | 2.20 | 0.13 | -187 | 130 | OK |
| 438 | NCL-4213076 | 18.449 | -92.757 | 7752 | 0.92 | -0.32 | 26.2 | 25 | 5 | 0.52 | 0.05 | 16.13 | 0.28 | 19.16 | 0.67 | 627 | 13 | 1.30 | 0.08 | 0.69 | 0.04 | 0.19 | 0.01 | 5.9 | 0.2 | 10 | 4 | 2.18 | 0.09 | 2.71 | 0.15 | -697 | 150 | OK |
| 436 | NCL-1114073 | 18.445 | -92.756 | 8199 | 2.2 | 0 | 24.8 | 25 | 5 | 0.87 | 0.09 | 15.72 | 0.23 | 18.32 | 0.53 | 643 | 13 | 1.30 | 0.08 | 0.68 | 0.04 | 0.15 | 0.01 | 5.4 | 0.2 | 12 | 3 | 2.14 | 0.09 | 2.98 | 0.17 | -967 | 170 | OK |
| 437 | NCL-4213075 | 18.442 | -92.751 | 8678 | 1.5 | 0 | 29.9 | 25 | 5 | 1.00 | 0.10 | 18.22 | 0.30 | 21.42 | 0.74 | 661 | 14 | 1.36 | 0.09 | 0.72 | 0.04 | 0.17 | 0.01 | 7.8 | 0.4 | 18 | 6 | 2.26 | 0.1 | 3.47 | 0.23 | -1457 | 230 | OK |
| 435 | NCL-1114074 | 18.436 | -92.751 | 9272 | 1.3 | -0.1 | 25.2 | 25 | 5 | 0.93 | 0.09 | 18.03 | 0.28 | 20.24 | 0.66 | 679 | 14 | 1.36 | 0.09 | 0.69 | 0.04 | 0.18 | 0.01 | 7.8 | 0.4 | 16 | 5 | 2.25 | 0.1 | 3.46 | 0.23 | -1447 | 230 | OK |
| 434 | NCL-4213074 | 18.430 | -92.751 | 9953 | 2.7 | -0.1 | 25.8 | 25 | 5 | 1.05 | 0.11 | 14.56 | 0.25 | 16.65 | 0.68 | 658 | 14 | 1.31 | 0.09 | 0.66 | 0.04 | 0.14 | 0.01 | 7.8 | 0.4 | 23 | 4 | 2.13 | 0.09 | 3.67 | 0.25 | -1657 | 250 | Likely OK |
| 433 | NCL-1114075 | 18.426 | -92.750 | 10398 | 1.5 | -0.1 | 26.1 | 25 | 5 | 0.68 | 0.07 | 18.86 | 0.23 | 21.46 | 0.50 | 632 | 13 | 1.32 | 0.09 | 0.71 | 0.04 | 0.17 | 0.01 | 8.3 | 0.5 | 22 | 4 | 2.20 | 0.09 | 3.77 | 0.28 | -1757 | 280 | Likely OK |
| 432 | NCL-4213073 | 18.422 | -92.749 | 10866 | 1.1 | -0.1 | 25.1 | 25 | 5 | 0.25 | 0.03 | 14.78 | 0.26 | 16.49 | 0.66 | 594 | 13 | 1.21 | 0.08 | 0.63 | 0.03 | 0.18 | 0.01 | 8.7 | 0.5 | 23 | 4 | 2.03 | 0.09 | 4.27 | 0.33 | -2257 | 330 | Likely OK |
| 450 | NCL-4213080 | 18.390 | -92.805 | 12637 | 1.05 | 0 | 26.9 | 25 | 5 | 0.58 | 0.06 | 13.06 | 0.24 | 15.49 | 0.6: | 529 | 12 | 1.08 | 0.07 | 0.56 | 0.03 | 0.18 | 0.01 | 8.4 | 0.3 | 12 | 2 | 1.83 | 0.08 | 4.58 | 0.26 | -2567 | 260 | OK |
| 451 | NCL-4213081 | 18.390 | -92.806 | 12684 | 1.48 | 1.17 | 3.1 | 5 | 3 | 0.86 | 0.09 | 12.18 | 0.17 | 13.66 | 0.35 | 566 | 12 | 1.37 | 0.08 | 0.66 | 0.03 | 0.18 | 0.01 | 8.8 | 0.3 | 15 | 2 | 2.23 | 0.08 | 3.97 | 0.21 | -1957 | 210 | OK |
| 452 | NCL-1114076 | 18.379 | -92.771 | 14412 | 1.6 | -0.1 | 24.5 | 25 | 5 | 0.63 | 0.06 | 14.69 | 0.17 | 16.62 | 0.33 | 595 | 12 | 1.21 | 0.08 | 0.66 | 0.03 | 0.16 | 0.01 | 9.8 | 0.4 | 22 | 3 | 2.00 | 0.08 | 4.91 | 0.3 | -2897 | 300 | Likely OK |
| 459 | NCL-4213082 | 18.420 | -92.994 | | 0.7 | -0.1 | 22.0 | 25 | 5 | 0.58 | 0.06 | 14.64 | 0.27 | 18.95 | 0.70 | 540 | 12 | 1.13 | 0.07 | 0.61 | 0.03 | 0.19 | 0.01 | 2.73 | 0.13 | 11 | 3 | 1.94 | 0.08 | 1.41 | 0.09 | 603 | 90 | OK |

1) composite distance from current coastline (m), projected along transect B (Fig. 3b)
2) the bootstrapped version of the Central Age Model (Cunningham and Wallinga, 2012) was applied to determine the burial dose of the samples.
3) for this sample the bootstrapped version of the Minimum Age Model (Cunningham and Wallinga, 2012) was applied to determine the burial dose. As over-dispersion input value (sigma_b) 18 ± 6% was used.
alpha dose rate of 0.010 ± 0.005 assumed for all samples

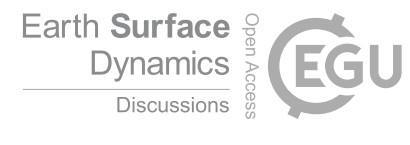

Table A3

| core | depth (m) | n | SiO$_2$ (%) | ± | TiO$_2$ (%) | ± | Al$_2$O$_3$ (%) | ± | FeO (%) | ± | MnO (%) | ± | MgO (%) | ± | CaO (%) | ± | Na$_2$O (%) | ± | K$_2$O (%) | ± | P$_2$O$_5$ (%) | ± | S (%) | ± | Cl (%) | ± | total (%) | before norm. (%) |
|---|---|---|---|---|---|---|---|---|---|---|---|---|---|---|---|---|---|---|---|---|---|---|---|---|---|---|---|---|
| 197 | 0.8 | 10 | 78.87 | 0.40 | 0.10 | 0.02 | 12.28 | 0.17 | 0.56 | 0.10 | 0.08 | 0.04 | 0.09 | 0.02 | 0.57 | 0.10 | 3.02 | 0.42 | 4.27 | 0.35 | 0.01 | 0.02 | 0.01 | 0.01 | 0.12 | 0.01 | 100 | 97.34 |
| 193[1] | 5 | 7 | 78.72 | 0.30 | 0.10 | 0.02 | 12.36 | 0.10 | 0.55 | 0.04 | 0.06 | 0.06 | 0.09 | 0.01 | 0.60 | 0.03 | 3.19 | 0.21 | 4.19 | 0.09 | 0.01 | 0.01 | 0.01 | 0.02 | 0.11 | 0.01 | 100 | 98.09 |
| 252 | 4.4 | 14 | 78.52 | 0.38 | 0.11 | 0.03 | 12.24 | 0.17 | 0.69 | 0.24 | 0.06 | 0.04 | 0.09 | 0.05 | 0.60 | 0.17 | 3.38 | 0.12 | 4.18 | 0.48 | 0.01 | 0.02 | 0.01 | 0.02 | 0.12 | 0.01 | 100 | 98.07 |
| 252 | 4.9 | 10 | 79.00 | 0.19 | 0.10 | 0.02 | 12.19 | 0.17 | 0.52 | 0.06 | 0.06 | 0.05 | 0.09 | 0.01 | 0.61 | 0.04 | 3.10 | 0.20 | 4.20 | 0.15 | 0.01 | 0.01 | 0.02 | 0.03 | 0.11 | 0.01 | 100 | 97.51 |
| 336 | 3.2 | 13 | 78.56 | 0.31 | 0.10 | 0.01 | 12.15 | 0.13 | 0.64 | 0.19 | 0.07 | 0.03 | 0.08 | 0.01 | 0.55 | 0.08 | 3.30 | 0.14 | 4.39 | 0.22 | 0.01 | 0.02 | 0.02 | 0.02 | 0.12 | 0.03 | 100 | 98.41 |

| | n | SiO$_2$ (%) | ± | TiO$_2$ (%) | ± | Al$_2$O$_3$ (%) | ± | FeO (%) | ± | MnO (%) | ± | MgO (%) | ± | CaO (%) | ± | Na$_2$O (%) | ± | K$_2$O (%) | ± | BaO (%) | ± | NiO (%) | ± | Cr$_2$O$_3$ (%) | ± | total (%) | before norm. (%) |
|---|---|---|---|---|---|---|---|---|---|---|---|---|---|---|---|---|---|---|---|---|---|---|---|---|---|---|---|
| TB III[2] | 14 | 78.83 | 0.49 | 0.10 | 0.01 | 12.84 | 0.37 | 0.58 | 0.05 | 0.07 | 0.02 | 0.08 | 0.01 | 0.59 | 0.04 | 2.72 | 0.39 | 4.09 | 0.22 | 0.11 | 0.04 | 0.01 | 0.02 | 0.00 | 0.00 | 100 | 97.98 | 2.57 |

1. pumice clast of 1.5 cm diameter
2. Tierra Blanca (Cabadas-Báez et al., 2017)



# Appendix B



Figure B1





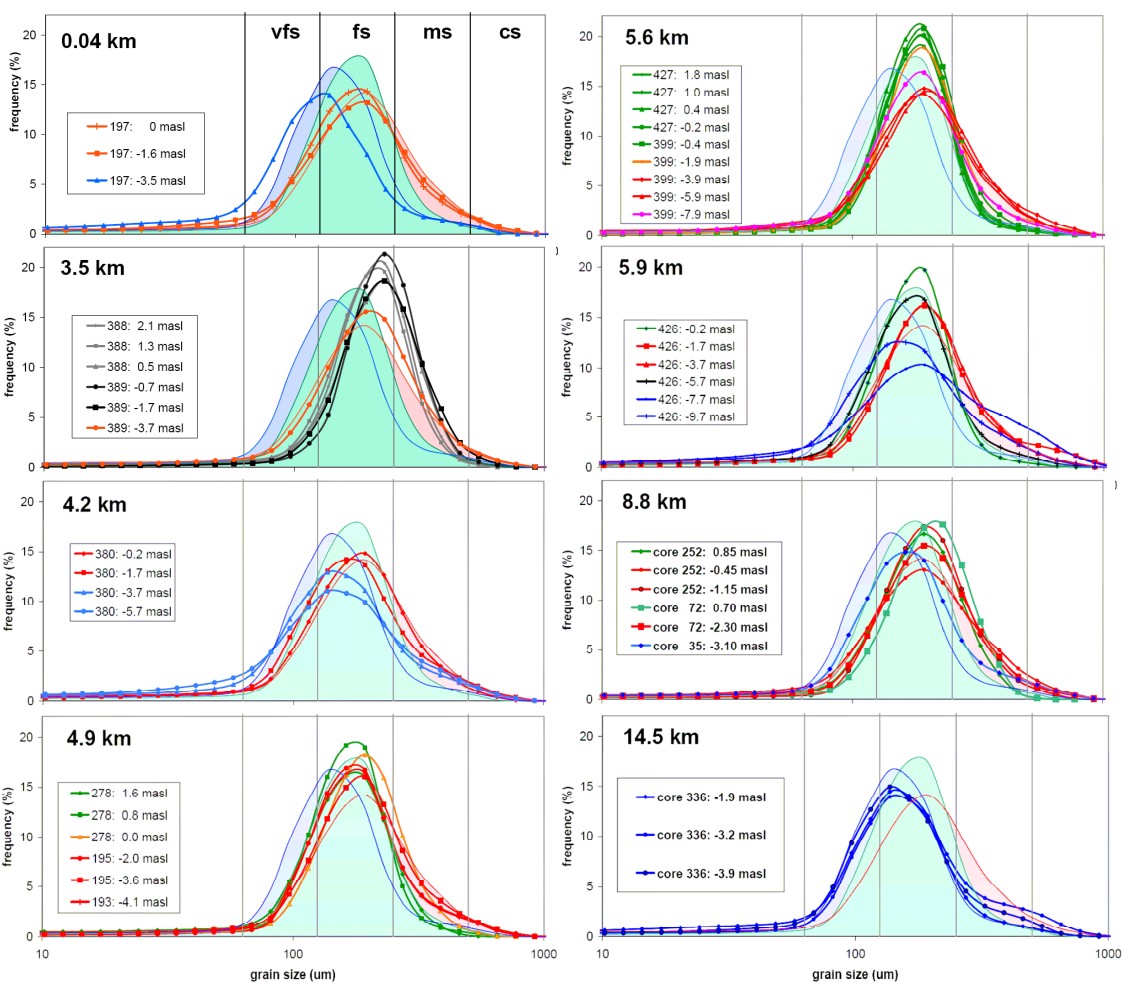

Figure B2