# Peer review of "The Usumacinta-Grijalva beach-ridge plain in southern Mexico: a high-resolution archive of river discharge and precipitation"

_Earth Surface Dynamics, 2017_

## Referee Comment (RC1) · T. Tamura (Referee) · 6 Jun 2017

I enjoyed reading this paper. It presents a laborious research that provided the re-fined chronology and morphological architecture of a huge coastal plain of Usumacinta-Grijalva in Mexico. While I would prefer to call this coastal system a wave-dominated delta rather than a beach-ridge complex, the paper presents valuable data which po-tentially provides insights into the nature of the beach ridge and associated long-term coastal evolution. You attempted making the paper focused and appealing by stress-ing the possible linkage of the beach-ridge height to the fluvial sediment supply, but the present data so far may not be convincing enough. With a nice combination of

[Figure]

OSL and radiocarbon chronology and Lidar elevation data, you found an interesting, negative correlation between the apparent progradation rate and the thickness of the aeolian accumulation on top of the prograded beach deposits, the core of beach ridge. The apparent progradation rate, however, in contrast to the title, is not related exclusively to the fluvial sediment supply, but also affected by several factors, such as the onshore wind strength and direction, and river avulsion and associated re-organization of the distributary channels. You are aware of this in the discussion. I also think that the river does not have to be an exclusive sediment source, as in the Gulf of Mexico, it is widely considered that the shelf sand is an important source of beach-ridge sands (e.g., Taylor and Stone, 1996, Journal of Coastal Research). You try to emphasize that the river has the large capacity of sediment supply with the aid of fragile Late Pleistocene volcanic rocks. However, I am curious where did the quartz grains, which are believed to comprise 50-65% of the beach ridge sand; L118-119), come from? I think they are unlikely from the volcanic rocks as volcanic quartz generally has inappropriate OSL properties for dating (e.g., Tsukamoto et al., 2003, Radiation Measurements). There should be a significant contribution of shelf sands as well, and thus the implied linkage between the beach-ridge height and fluvial discharge (as in the title) would be misleading. I suggest you reorganize the paper as just to examine the relationships between the ridge development and the apparent progradation, properly address the presence of other sediment sources, and then thoroughly discuss how the coast developed. Apart from this, I found numerous minor and moderate points that I hope you to reconsider as follows. In summary, I support the publication of this paper, but there is much to be reconsidered and refined.

Title: This coastal system looks like a wave-dominated delta rather than a beach-ridge complex. It would be evident that the fluvial sediment supply is significant if in the delta system. There is yet a large gap until you can use the beach ridge height as proxies of fluvial discharge and precipitation. In this paper you do not say anything about the temporal fluctuations in precipitation in the catchment, and so the title would be misleading. L81: insert Usumacinta-Grijalva before 'beach-ridge plain'. L85-86:

Why not cite the criticism by Tanner & Stapor (1971, Trans. Gulf Coast Assoc. Geol. Soc.). L117-127: This argument should be shortened and merged into the last one. L149: 'the same extent', bigger or smaller? L164-165: insert 'high' after '0.3 to 0.7 m'. L185-192: These arguments are more than the methodology. L265: 90Sr/90Y beta source L276-279: This argument is not necessary or accurate, and should be removed. L301-302: This does not make sense here before knowing the details of the dataset. L304-309: This section needs more detail. L312: should shore-normal, nor cross-normal. L319-320: You need core logs presented in the paper or at least in the appendix to estimate the thickness of the deposits. You may be confused with what is 'beach-ridge deposits'. It should not be so thick here, and does not include the subtidal shoreface deposits. Barrier sand or barrier deposits would be a broader term appropriate for what you call 'beach-ridge deposits'. L327-328: I am not convinced with your assumption of the constant run-up height. The boundary between beach/foredune could be the swash limit of the higher waves during the spring high tide. The spring tide range is 0.75 m and the swells can be up to 0.7 m high, yielding higher swash height than +0.5 m above the mean sea level. Besides, the NW swells can be up to 1.7 m high, implying the longshore gradient of swash height. L346: There appear to be two sets of definition of transects (A-C, and 1-15). This is confusing. L354: To me the faults look almost 45 degrees to the shore. L355: Not clear whether it means eastward tilting or westward tilting. L358: I cannot find any holes in Fig. 2b. L394-396: This may repeat the argument in the methodology, and strictly, is a bit different from that. Above you appear to have said the MAM was applied to all cases with OD >30%. L399-403: Delete. L410: The significance of the 600 years depends on the absolute age and so you should mention the absolute values of these two ages here. L424-426: Remove this argument, as it is not necessary. L454-461: I do not like so much discussion and interpretation presented in the result section. The last argument is problematic; how can you quantify the aeolian accretion rate separately from the chronology? L469-470: Remove this. L495: Be sure that the steepening of the shoreface is related to the coarser sand but not to the higher waves. L502: This is a microtidal system and

tidal current is minor. L520-521: I cannot follow this. It is too obvious to say that some aeolian processes act on the backshore and foredune ridges. L550-552: This sounds over-interpretation. Where is the swash bar welded to the beach corresponding the infill of a large runnel? L562-563: This is where you mention Tanner and Stapor (1971). L601: You should describe how you picked up the general trend. You appear to pick up the bottom of the trough/swale; is there rational behind this? L619-621: How did you define the thickness of what you call beach-ridge deposits at Keppel and Guichen bays, while in these sites the subtidal barrier deposits have not be explored. L636: This number highly depends on the thickness of the subtidal barrier deposits. L641: Why? Transect B occurs across the NE-SW to ENE-WSW trending coast, and the easterly wind blows from behind the foredune ridge and should not promote the onshore aeolian accumulation. L651: Fig. 3b is not informative enough. You need to show core logs at least in the appendix. L690-692: How can you tell the shape of the eroded, lost delta river mouth? Is it evident from Fig. 2a? L696: the marked increase is not evident in Transect A. L701-703: What data critically show this? L709: The break is not evident in figures you provide. There is no apparent age gap as well. How can you define it? L718-720: A similar view was given by Tanner and Stapor (1971). L739: cite Shepherd (1991, in Applied Quaternary Studies. Australian National University). L755: But other periods show more prominent ridges. L784: This is just what you imagine. How certain is it? L792: Hopefully you show some independent climate records here. Fig. 1b: Where did the sediment come from to form the updrift (eastern) part of the Phase 3 just next to Ciudad del Carmen. Fig. 7: What do you mean by 'beach/ridge'? 'Beach' is a broad term including the swash zone as well. Fig. 12: Why you show multiple plots for each episode of transect. Just one mean value for each is fine.

---

## Referee Comment (RC2) · E. Otvos (Referee) · 8 Jun 2017

This an excellent, well-researched paper with a huge volume of sound research data and a wealth of appropriate references. Excellent illustrations. Three major points of criticism: (1) The value used for ancient sea level position, foreshore-eolian interface elevation above MSL (0.5 m) seems to be arbitrary. It is not clearly shown how the MSL is independently established in a given drillcore section inland from the present Gulf shore. How was MSL identified in a given drillhole before plotting on the cross sections? (2) No indication that an attempt was made to use/show textural statistical data, ratios between (utilizing kurtosis, skewness, sorting, etc.) of sand samples to dis-

tinguish, draw the horizontal boundary between the facies. There have been numerous papers on the subject in the literature. At least those should be used to show at least why such a potentially promising approach does not work. (3) The paper reject the idea L.6that tropical cyclones or other processes play/played any role (L. 684-688) in (occasionally, under favorable conditions very significant) landward sand transfer from shelf sources across the inner shelf. Some of these sands may have been first swept out from shore/nearshore sites, then swept back by the same or another cyclone. Certainly, this must/may have happened on the cited shore sector as well. Non-storm fair weather landward sand migration from offshore, inner shelf or nearshore sources is, additionally shown by their cited landward shifting then welding swash bars. Apart from this, only the erosional aspects of hurricanes are noted - even then you should have written briefly something about local hurricane history, recurrence, magnitude, years of major impact, etc.

A few typos, style errors: Line 72 instead of "strong reduction...etc." write: significant slowing/deceleration (or reduction) in the rate of sea level rise L.86 played important role "to"(?)... "in" would be better L. 409 it is not Sint, but Saint George or St. George Island L. 251 "hardly harmed" sounds funny and not geological, write something like angular, non-rounded, with irregular outline

---

## Author Comment (AC1) · 15 Jun 2017

We thank the two reviewers for their constructive comments on our paper "The Usumacinta-Grijalva beach-ridge plain in southern Mexico: a high-resolution archive of river discharge and precipitation". We agree with most of the issues raised. They will be addressed in the final version of the paper.

Here, we only give additional information related to the main points.

First of all, our manuscript focuses on a particular portion of the Usumacinta-Grijalva beach-ridge complex that is and has always been very close to main river mouths

of this double river system. Maybe we should more specifically call them 'river-mouth beach-ridge plains' or 'promontory' portions of the UG-delta In the paper we use Otvos' (2000) broad definition of beach ridges (L75-78), and include all Holocene swash- and aeolian deposits within the beach-ridge plain deposits. The term "wave-dominated" (cf. Galloway, 1975) does indeed apply to the barrier-coastline morphology of the UG-delta as a whole (Figure 1a; manuscript). We could include this term when revising the introduction section, but in our opinion it is a too broad and general term to characterize the specific river-mouth beach ridge amalgamated promontory complex that is the object of study in the paper (and we purposely avoided using it so far).

RC1's main point is that he is not convinced that Holocene fluvial sediment supply is the dominant sediment source for the construction of the beach-ridge plain. He refers to studies along the Northern Gulf of Mexico coast where shelf sand is considered an important sediment source. Also RC2 mentions a possible redistribution of shelf sand, connected to the role of tropical storms and hurricanes which likely played an important role along the Northern Gulf of Mexico coast. It should be realized that coastal wind and wave conditions in the Northern Gulf of Mexico are remarkably different then at the relatively protected southern Gulf coast. Hurricanes are a frequent phenomenon in the Gulf of Mexico (e.g., Kossin et al., 2010), but they generally pass over the middle and northern part, whereas landfall at or near the study site is rare (www.nhc.noaa.gov/data/#tracks_all). One should also realize that the beach-ridge plains along the Northern Gulf of Mexico coast are small compared to that of our study area (Fig. 1; this reply).

We agree that shelf transport adds to beach barrier plain build out and beach-ridge formation, and that it does so along the entire length of the UG-delta coast line (at the scale that wave-dominated delta terminology applies). We stress, however, that along the river mouths the beach-ridges accrete and build out at significantly larger rates and we postulate that at these proximal positions, it is the Holocene fluvial sediment supply that majorly adds to the shelf sand supply – both to subtidal (shoreface, foreshore)

as well as to intertidal and supratidal (swash and aeolian processes). After fluvial processes delivered the sediment to the shallow sea, we agree with the reviewers that coastal wind and wave conditions take over and work up sediments (which in the direct vicinity of the river mouth we regard a mix of younger fluvial deliver and older shelf-derived components) to form the progradational beach-ridge series, most efficiently and at highest resolution closest to the river mouth.

In general, shelf sand in the near-shore zone should be regarded a temporary sediment storage to supply sand for beach-ridge construction and coastal progradation. Under conditions of late Holocene relative sea-level rise, this storage required a continuous long-term sediment supply from an indirect source. Few beach-ridge studies consider such an indirect sand source, including most studies cited by Taylor and Stone (1996); the work of Brooke et al. (2008) is one of the few exceptions. They demonstrated that 79% of the estimated long-term average annual bedload of the Fitzroy River, temporally stored in Keppel Bay, likely contributed to beach-ridge plain formation at Long Beach, Australia. In a recent study, Mammi et al. (2017) demonstrated fluvial input to be the main source of beach ridges at the Ombrone River delta (Italy), and that variations in beach ridge elevation are due to variations in fluvial sediment supply.

We do not wish to exclude non-fluvial sandy sediment sources, but multiple arguments have been presented (L676-694) that, during beach-ridge formation phase 2, the Usumacinta River was likely the main source of sediment for the construction of the old SP y SP promontory. Although we believe that weathered Los Chocoyos ignimbrites are an important sandy sediment source (L569-594), RC1 suggests that it is unlikely that quartz grains are from weathered volcanic rock, because volcanic quartz from certain types of volcanoes has inappropriate OSL properties for dating (e.g. Tsukamoto et al., 2003). However, Pietsch et al. (2008) demonstrated that OSL sensitivity of quartz increases linearly with fluvial transport distance for the Castlereagh River in Australia. Such a sensitization effect might explain the decent OSL sensitivity of our samples, even if they started as ignimbrites with poor OSL sensitivity in the upper catchment

(∼1100 km). Another explanation could be sought in a secondary source of quartz with high luminescence sensitivity. Even if the bulk of the sediment is from ignimbrites, a minor component from another source may be responsible for the observed OSL signal.

We understand RC1's and RC2's point (1) that the interface between the aeolian and swash facies seems somewhat arbitrarily set at a constant height of 0.5 m above the mean sea level at the time of deposition. Indeed, given the temporal and spatial variability in run-up height, the uncertainties in the absolute elevation of beach ridge sand samples, late Holocene estimated RSL rise, and the limited number of grain size data, defining it as a bandwidth of $0.5 \pm 0.5$ would be more appropriate. We will correct this in the paper and will mention that our calculated aeolian accretion rates are indicative only.

We agree with point (2) of RC2 on the limited presence of descriptive statistics on grain size data in the paper. We will repair this and will add two additional figures (Figs. 2 and 3; this reply).

The grain size statistics (Fig. 3.; cf. Martins, 2003) indicate that aeolian beach and dune sands can be distinguished from the swash sands by a better sorting and an increased skewness (more symmetrically skewed). Sand deposited during phase 3B (Fig. 2) has similar sorting and skewness characteristics but has a coarser mean grain size (smaller (phi)), due to the contribution of reworked sand from the eroding SP y SP promontory.

RC1 is correct that the scour holes cannot be recognized in Fig. 2. We will therefore include a new figure (Fig. 4) to illustrate the presence of scour holes along the beach ridge formed around 1450 CE.

RC1 misses core logs. However because sediment lithology is very uniform at all core locations, and sedimentary microstructures not recovered in Van der Staay hand corings, there is little need to present them.
RC1 misses an extended discussion on river discharge, precipitation and climate change. In view of the already considerable length of the paper, we prefer to discuss these subjects elsewhere (Nooren et al., in prep.).

References

Blott, S.J, and Pye, K.: Gradistat: A grain size distribution and statistics package for the analysis of unconsolidated sediments, Earth Surface Processes and Landforms 26, 1237-1248, 2001.

Brooke, B., Ryan, D., Pietsch, T., Olley, J., Douglas, G., Packett, R., Radke, L., and Flood, P.: Influence of climate fluctuations and changes in catchment land use on Late Holocene and modern beach-ridge sedimentation on a tropical macrotidal coast: Keppel Bay, Queensland, Australia, Marine Geology, 251, 195–208, 2008.

Forrest, B.M.: Evolution of the Beach Ridge Strandplain on St. Vincent Island, Florida, Thesis, Florida State University, 269 pp., 2007.

Galloway, W.E.: Process framework for describing the morphologic and stratigraphic evolution of deltaic depositional systems. In: Broussard, M.L. (Ed.), Deltas, Models for Exploration, Houston Geological Society, Houston, TX, 87-98, 1995.

Kossin, J.P., Camargo, S.J., and Sitkowski, M.: Climate modulation of North Atlantic Hurricane tracks, Journal of Climate 23, 3057-3076, 2010.

López, G.I., and Rink, W.J.: New quartz optical stimulated luminescence ages for beach ridges on the St. Vincent Island Holocene strand plain, Florida, United States. Journal of Coastal Research, 24, 49–62, 2008.

Mammi, I. Piccardi, M., Pranzini, E. and Rossi L.: Reading Ombrone river delta evolution through beach ridges morphology. EGU General Assembly 2017; Geophysical Research Abstracts, Vol. 19, EGU2017-4010-1, 2017.

Martins, L.R.: Recent sediments and grain-size analysis, Gravel 1, 90-105, 2003.

Nooren, K., Hoek, W.Z., Dermody, B.J., Galop, D., Malaizé, B., Metcalfe, S., Islebe, G. and Middelkoop: Climate impact on the development of Pre-Classic Maya civilization, in prep.

Otvos, E.G.: Beach ridges — definitions and significance, Geomorphology, 32, 83–108, 2000.

Otvos, E.G.: Coastal barriers, Gulf of Mexico: Holocene evolution and chronology, Journal of Coastal Research, SI(42), 141-163, 2005.

Pietsch, T.J., Olley, J.M., and Nanson, G.C.: Fluvial transport as a natural luminescence sensitiser of quartz, Quaternary Geochronology 3, 365-376, 2008.

Rink, W.J. and López, G.I.: OSL-based lateral progradation and aeolian sediment accumulation rates for the Apalachicola Barrier Island Complex, North Gulf of Mexico, Florida, Geomorphology, 123, 330-342, 2010.

Tanner, W.F.: Late Holocene sea-level changes from grain-size data: evidence from the Gulf of Mexico, The Holocene, 2, 249–254, 1992.

Taylor, M.J., and Stone, G.W.: Beach-ridges: a review, Journal of Coastal Research, 12, 612–621, 1996.

Tsukamoto, S., Rink, W.J., and Watanuki, T.: OSL of tephric loess and volcanic quartz in Japan and an alternative procedure for estimating De from a fast OSL component, Radiation measurements 37, 459-465, 2003

**Fig. 1.** Two large beach-ridge plains along the GOM coast. Reconstructed palaeocoastlines (upper panel) after Tanner (1992), López and Rink (2008), Rink and López (2010), Otvos (2005), and Forrest (2007).

[Figure]

**Fig. 2.** : Sand samples along transect A for which grain size parameters are indicated in Fig. 3. See Fig. 3 for explanation of symbols.

[Figure]

**Fig. 3.** Grain size statistical parameters, calculated conform the logarithmic method of moments (Blott and Pye, 1975).

**Fig. 4.** Scour holes along a beach ridge formed around 1450 CE.